# Artesunate treats obesity in male mice and non-human primates through GDF15/GFRAL signalling axis

Xuanming Guo [1,8], Pallavi Asthana[1,8] ✉, Lixiang Zhai[1], Ka Wing Cheng [1,2], Susma Gurung[1], Jiangang Huang[3], Jiayan Wu[1], Yijing Zhang[1], Arun Kumar Mahato [4], Mart Saarma [4], Mart Ustav[5], Hiu Yee Kwan [1], Aiping Lyu [1], Kui Ming Chan [6], Pingyi Xu[7], Zhao-Xiang Bian [1,2] ✉ & Hoi Leong Xavier Wong [1] ✉

Obesity, a global health challenge, is a major risk factor for multiple life-threatening diseases, including diabetes, fatty liver, and cancer. There is an ongoing need to identify safe and tolerable therapeutics for obesity management. Herein, we show that treatment with artesunate, an artemisinin derivative approved by the FDA for the treatment of severe malaria, effectively reduces body weight and improves metabolic profiles in preclinical models of obesity, including male mice with overnutrition-induced obesity and male cynomolgus macaques with spontaneous obesity, without inducing nausea and malaise. Artesunate promotes weight loss and reduces food intake in obese mice and cynomolgus macaques by increasing circulating levels of Growth Differentiation Factor 15 (GDF15), an appetite-regulating hormone with a brainstem-restricted receptor, the GDNF family receptor α-like (GFRAL). Mechanistically, artesunate induces the expression of GDF15 in multiple organs, especially the liver, in mice through a C/EBP homologous protein (CHOP)-directed integrated stress response. Inhibition of GDF15/GFRAL signalling by genetic ablation of GFRAL or tissue-specific knockdown of GDF15 abrogates the anti-obesity effect of artesunate in mice with diet-induced obesity, suggesting that artesunate controls bodyweight and appetite in a GDF15/GFRAL signalling-dependent manner. These data highlight the therapeutic benefits of artesunate in the treatment of obesity and related comorbidities.

Excess weight, especially obesity, has been found to increase the risk of various severe and even life-threatening diseases including cardiovascular diseases, and some cancers[1]. Obesity induces insulin resistance, a well-known factor for Type 2 diabetes (T2D), and it is estimated that obesity-related diabetic population will reach 300 million by 2025[2–5]. The worldwide increase in the incidence of obesity has emerged as a major threat to public health, imposing tremendous economic burdens and medical challenges to society. To reduce the burden on public healthcare costs caused by obesity and its comorbid conditions, it is important to identify new therapeutic approaches and agents that can safely and effectively manage the body weight of people with obesity.

Currently, treatment of obesity mainly involves a combination of lifestyle changes and medications, although their effectiveness can

vary, and whilst bariatric surgery is deemed the most effective treatment of obesity, it nonetheless comes with risks associated with major surgery. Furthermore, although exercise and dietary changes have been found to be effective in reducing body fat and alleviating metabolic dysfunction, these strategies are also hampered by issues of patient compliance. Nonetheless, a wide range of therapeutics exist for the continuous management of obesity, including orlistat, lorcaserin, liraglutide, and combinations of phentermine/topiramate or bupropion/naltrexone[6]. However, a large proportion of these drugs offer modest effects on weight reduction and are associated with adverse effects, such as cardiovascular complications, suicidal risk effects, and enhanced likelihood of drug dependence, resulting in the withdrawal of multiple anti-obesity drugs from the market after regulatory approval[7]. The market for anti-obesity drugs is facing a new phase with the introduction of new weight-loss drugs, such as tirzepatide, semaglutide, and orforglipron. Although the adverse effects listed in association with current therapies are not critical issues for tirzepatide, semaglutide, and orforglipron, clinical trials have revealed that a quarter of participants taking one of these drugs reported that they discontinued the treatments due to gastrointestinal adverse events[8,9]. In addition to incretin-based therapeutics, the hormone leptin is thought to be a driving force for the treatment of obesity through reducing food intake and increasing energy expenditure[10–14]. However, leptin-based therapeutics in clinical trials have failed to effectively treat obesity, predominately owing to leptin resistance in most people with obesity[15–19]. Fibroblast growth factor 21 (FGF21) is also emerging as a key stress-induced hormone in the regulation of energy homeostasis. However, the clinical implementation of FGF21-based therapies for the treatment of obesity is restricted by the complexity of pathophysiology of FGF21, the potential development of FGF21 resistance in obesity and the poor pharmacokinetic of native FGF21[20–24]. Therefore, there is an ongoing need to develop pharmacological and non-invasive therapeutics with suitable tolerability for the management of obesity.

Brain regions, particularly hypothalamus and hindbrain, comprise a network of neural circuits which regulate energy homeostasis and body weight in response to metabolic, gastrointestinal, and endocrine signals[10,11]. These neural networks are regulated by diverse appetite-regulatory hormones in which growth differentiation factor 15 (GDF15) has been identified as one of the key players in multiple metabolic disorders, such as obesity and diabetes[25–27]. Loss of *Gdf15* leads to increased weight gain[28]. Conversely, increasing GDF15 levels by transgenic overexpression or pharmacological administration of recombinant GDF15 induces weight loss and improves cardiometabolic parameters in mice with diet-induced obesity and macaques with spontaneous obesity[26]. GDNF family receptor α-like (GFRAL), strictly expressed in the brainstem area postrema and nucleus of the solitary tract, has been identified as a specific receptor for GDF15[29–32]. Depletion of GFRAL abrogates the anorexic effects of GDF15[29–32], revealing the importance of GFRAL for the GDF15 mechanism of action. Given the appetite-regulatory nature of GDF15, it has drawn much attention from the basic research and pharmacological sectors for its potential as an anti-obesity agent. However, pathological levels of GDF15 induce nausea and emesis[33], which is implicated in cancer cachexia and chemotherapy-induced anorexia. Therefore, identifying a pharmacological molecule that moderately induces GDF15 within tolerable levels is crucial for establishing GDF15-targeted therapies in a safe and efficacious manner.

In this study, we identified artesunate as a safe and effective therapeutic agent for the prevention and treatment of obesity in both primates and non-primates at tolerated doses. Artesunate, an artemisinin derivative approved by the FDA for the treatment of severe malaria, is well-tolerated in healthy individuals. This drug is safe for use in children, pregnant women in the second and third trimesters, and during lactation. The most common side effects of artesunate are nausea and dizziness, but these effects are mostly associated with acute malaria in many patients[34]. More serious complications associated with the use of artesunate, including neutropenia, anaemia, and haemolysis, have been found to be very rare, with an estimated risk of approximately 1 reaction per 3000 treatments[35]. Again, these effects are likely attributable to severe malaria in the patients rather than to the drug[35]. In this study, we found that artesunate treatment at tolerated doses reduced food intake, promoted weight loss, and ameliorated metabolic dysfunctions in both mice with high-fat diet-induced obesity and cynomolgus macaques with spontaneous obesity without the induction of nausea and malaise as well as pathological changes. We further demonstrated that the anti-obesity effect of artesunate was mainly mediated by the GDF15/GFRAL signalling axis. Our results highlighted the therapeutic potential of artesunate as a repurposed drug for the prevention and treatment of obesity.

## Results

### Artesunate ameliorates obesity and metabolic dysfunctions in mice

Previous studies have demonstrated the beneficial effects of artemisinin derivatives in reducing body weight in mice with diet-induced obesity[36]. However, the mechanism of action of artesunate remains unclear. To characterise the anti-obesity effect of artesunate, we examined the chronic consequences of artesunate treatment by treating wild-type C57BL/6 mice fed a high-fat diet (HFD) with intraperitoneal administration of artesunate. In wild-type mice fed the control diet, artesunate treatment did not significantly alter body weight (Fig. S1). However, in HFD-fed mice, chronic artesunate treatment reduced food intake and suppressed weight gain over time (Fig. 1A, B). Consistently, the fat mass of artesunate-treated mice was significantly lower than that of the vehicle-treated controls (Fig. 1C), suggesting that artesunate prevented the development of obesity. Importantly, reduced weight gain in obese mice after artesunate treatment was associated with lower fasting glucose and insulin levels, as well as improved glucose tolerance (Fig. S2A–D). Insulin levels were also considerably lower in artesunate-treated mice during glucose challenge (Fig. S2E), indicating enhanced insulin sensitivity. In line with the enhanced glucose excursion during the glucose challenge, the insulin sensitivity of artesunate-treated mice was significantly enhanced in the insulin tolerance test (Fig. S2F, G). In addition, artesunate treatment suppressed hepatocellular fat accumulation and alleviated the destruction of hepatic architecture in mice fed with HFD (Fig. S3A–F), which indicates that artesunate treatment prevented hepatic steatosis in mice challenged with a high-fat diet. To further explore the therapeutic effect of artesunate in the treatment of obesity, we administered artesunate to mice with diet-induced obesity. Daily administration of artesunate over a period of 13 days led to a reduction in body weight of approximately 10% in obese mice, which was attributable to lower food intake (Fig. 1D, E). As a result, the body fat of obese mice was dramatically reduced by approximately 30% (Fig. 1F). Furthermore, pair-feeding wild-type mice with an amount of food matching that consumed by mice treated with artesunate resulted in equivalent weight loss and reduction of fat weight between the treatments (Fig. 1G, H), indicating that the weight-loss benefits conferred by artesunate treatment are mainly attributed to the reduced energy intake. Compared with equimolar doses of other well-characterised weight loss drugs, including metformin (MEF) and liraglutide (LIR), treatment with artesunate showed superior potencies in terms of body weight regulation (Fig. S4A, B), food intake control (Fig. S4C), insulin sensitisation (Fig S4D–H), adiposity reduction (Fig. S5A–C), reduction of cholesterol levels in serum (Fig. S5D–G), and alleviation of hepatic steatosis (Fig. S6A–F) in mice with diet-induced obesity. These results collectively suggested that artesunate normalised body weight and improved metabolic profiles in obese mice, both prophylactically and therapeutically.

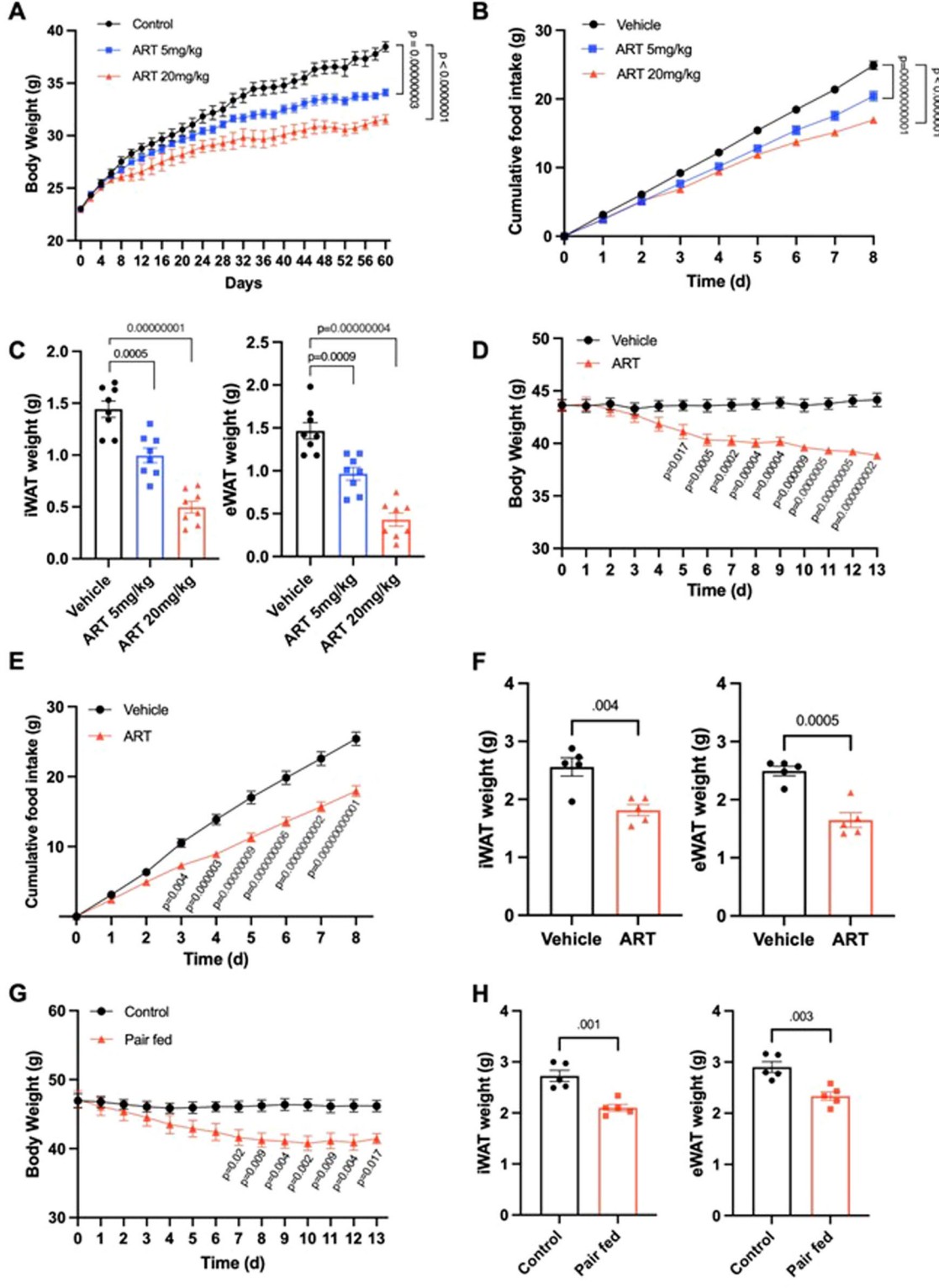

**Fig. 1 | Artesunate induces weight loss in obese mice by suppression of food intake. A–C** Wild-type mice were treated with vehicle or artesunate (ART; 5 mg/kg or 20 mg/kg) during high-fat diet (HFD) challenge for 60 days **A** Changes in body weight of mice during HFD feeding **B** 8-day cumulative food intake of mice measured at week 8 upon HFD challenge **C** weights of inguinal white adipose tissue (iWAT) and epididymal white adipose tissue (eWAT) of mice at the end of HFD challenge (*n* = 8 per group) **D–F** Wild-type mice with diet-induced obesity (DIO) were therapeutically treated with vehicle or ART (20 mg/kg) for 13 days. **D** Changes in body weight of mice during the treatment **E** 8-day cumulative food intake of mice over the treatment period **F** weights of iWAT and eWAT in mice at the end of

treatment (*n* = 5 per group) **G, H** Pair-fed animals were provided with the amount of food equivalent to the average intake for the ART-treated mice (20 mg/kg). The control group was allowed ad libitum access to food. **G** Changes in body weight during pair-feeding caloric restriction. **H** weights of iWAT and eWAT in pair-fed mice and corresponding controls at the end of pair-feeding (*n* = 5 per group). Data are expressed as mean ± SEM; one-way ANOVA followed by Tukey's test (**C**), two-way ANOVA followed by Šídák multiple comparison post-hoc test (**A**, **B**; **D**; **E**; **G**) or two-tailed unpaired Student's *t*-test (**F**, **H**). Source data are provided as a Source Data file.

### Artesunate confers therapeutic effects against obesity in cynomolgus macaques

To determine whether the weight-loss benefits conferred by artesunate treatment translated to a higher species, we examined the response of a non-human-primate model to artesunate treatment. Non-diabetic cynomolgus macaques with spontaneous obesity receiving intravenous administration of artesunate every two days were monitored for two weeks (Fig. 2A). Macaques treated with artesunate had significantly less food intake than vehicle-treated controls, resulting in a reduction in cumulative food intake over the monitored period in artesunate-treated animals (Fig. 2B). Throughout the study, artesunate-treated animals exhibited significant declines in body weight without obvious rebound weight gain upon treatment discontinuation (Fig. 2C, D). Notably, fasting insulin and blood glucose

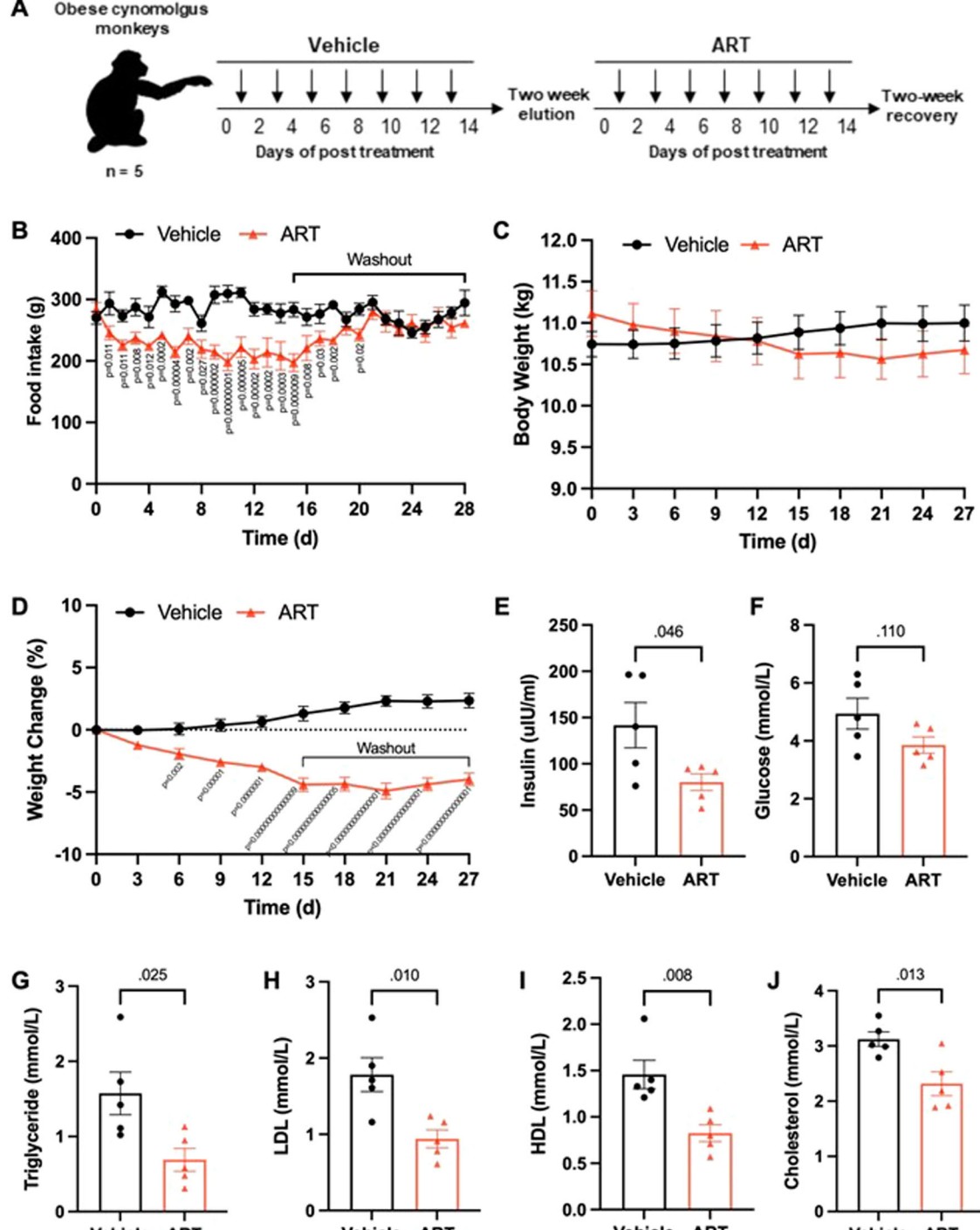

**Fig. 2 | Artesunate drives weight loss in non-human primates through food intake reduction. A** Schematic diagram of artesunate (ART) treatment in the obese cynomolgus monkey. **B–D** Daily food intake (**B**) and changes in body weight (**C, D**) in obese cynomolgus macaques treated with ART (6.4 mg/kg) or vehicle over the treatment period. **E–J** Levels of serum insulin (**E**), serum glucose (**F**), triglyceride (**G**), low-density lipoprotein (LDL) cholesterol (**H**), high-density lipoprotein (HDL) cholesterol (**I**), total cholesterol (**J**) in fasted obese cynomolgus macaques at the end of treatment. Data are presented as mean ± SEM, $n = 5$ per group; two-way ANOVA followed by Fisher's LSD post-hoc test (**B–D**) or two-tailed unpaired Student's $t$-test (**E–J**). Source data are provided as a Source Data file.

levels were lower in artesunate-treated animals (Fig. 2E, F), indicative of improved insulin sensitivity. Along with reduced levels of triglycerides, total cholesterol, LDL, and HDL (Fig. 2G–J), circulating levels of alanine aminotransferase (ALT) and aspartate aminotransferase (AST), two liver enzymes associated with hepatic damage, showed decreasing trends in artesunate-treated macaques (Fig. S7A, B). There was no artesunate-induced renal toxicity in obese macaques (Fig. S7C–F). Consistently, regular blood tests in obese non-human primates revealed that the 14-day treatment with artesunate resulted in significant improvements in both the hepatic and metabolic parameters without causing any major pathological changes (Table S1A, B). More importantly, clinical observations revealed no signs of emesis in macaques treated with artesunate. These data from non-human primates demonstrated that artesunate treatment effectively reduces body weight and reverses metabolic disturbances without inducing malaise, highlighting the therapeutic potential of artesunate repurposed for the treatment of obesity in humans.

### Artesunate is an inducer of GDF15

Given the fact that artesunate has been identified as an inducer of endoplasmic reticulum (ER) stress and *GDF15* is a stress response gene, we reasoned that the weight-loss benefit of artesunate was associated with elevated GDF15 production. We first examined the level of serum GDF15 in mice with diet-induced obesity and normal mice. Artesunate treatment induced an increase in serum GDF15 levels in mice fed the control diet; this inductive effect was further exaggerated in mice with diet-induced obesity (Fig. 3A). Consistently, the level of serum GDF15 was considerably elevated in obese macaques over 48 h after a single dose of artesunate (Fig. 3B). To identify the cellular source of GDF15 in response to artesunate treatment, we examined the expression of GDF15 in multiple organs reported to produce GDF15. Upon artesunate treatment, increased levels of GDF15 were observed in multiple organs, with the most significant upregulation in the liver (Fig. 3C). Consistently, western blotting analyses revealed elevated expression of GDF15 in the liver of mice upon artesunate treatment (Fig. 3D). To further confirm the direct effect of artesunate on GDF15 production, we examined the expression of GDF15 in MIHA cells, a normal human liver cell line, and primary mouse embryonic fibroblasts (MEFs). Artesunate increased GDF15 expression in a dose-dependent manner in both cell types (Fig. 3E, F; S8A). The results obtained from in vitro and in vivo studies suggested that artesunate induces GDF15.

### Artesunate-induced GDF15 production is mediated by the integrated stress response

Previous studies have reported that GDF15 expression is mainly regulated by the integrated stress response with CHOP as a primary mediator[37,38]. In alignment with previous reports that artesunate is an activator of the integrated stress response[39], artesunate increased the expression of CHOP concomitantly with GDF15 in mouse livers in vivo (Fig. 3D) and multiple cell lines including MIHA and MEFs in vitro (Fig. 3E, F; S8A). To determine whether the integrated stress response was involved in artesunate-induced GDF15 production, we inhibited the integrated stress response using pharmacological antagonists. We found that treatment with ISRIB, a potent inhibitor of the integrated stress response, effectively suppressed the artesunate-induced increase in CHOP and GDF15 expression in both MIHA and MEFs (Fig. 3G; S8B). Similarly, siRNA-mediated CHOP knockdown largely abolished the upregulation of GDF15 in response to artesunate treatment in Hep-G2 cells (Fig. 3H). These data demonstrated that artesunate promotes the production of GDF15 through CHOP.

### The GDF15/GFRAL axis mediates the anti-obesity effect of artesunate

To investigate the contribution of GDF15 to the anti-obesity effect of artesunate, we used an adeno-associated virus (AAV8) expressing *Gdf15*-specific shRNA (sh*GDF15*) to knock down GDF15 in the liver and kidney, from which artesunate-induced GDF15 is mainly derived (Fig S9A, B). As expected, artesunate-induced GDF15 production was dramatically reduced in DIO mice with sh*GDF15* (Fig. S9C). Despite no significant effect on energy expenditure (Fig. S10A, B), the suppression of food intake and subsequent weight loss by artesunate treatment were largely diminished in mice with sh*GDF15* (Fig. 4A, B). We found a moderate increase in physical activities in artesunate-treated mice, but this effect of artesunate was abolished by the knockdown of GDF15 (Fig. S10C, D). We next analysed respiratory exchange ratio (RER) in DIO mice and found artesunate moderately reduced RER irrespective of GDF15 levels, indicative of increased metabolism of fatty acids and reduced carbohydrate utilisation upon artesunate treatment (Fig. S10E, F). Moreover, the beneficial effects of artesunate on glucose tolerance and insulin sensitivity in DIO mice were remarkably reduced by sh*GDF15* treatment (Fig. 4C–F). To further confirm the results, we examined the activation of GFRAL neurons by using the neuronal activity marker c-Fos. We observed significant higher number of activated GFRAL neurons in the artesunate-treated sh*Control* mice, but this neuronal activation was abolished in sh*GDF15* mice (Fig. S11A, B). To further test whether GFRAL is essential for the weight reduction effect of artesunate, we examined the response of *Gfral*⁻ᐟ⁻ mice to artesunate treatment. Although artesunate elevated similar levels of serum GDF15 in *Gfral*⁻ᐟ⁻ mice and wild-type littermate controls (Fig. 5A), it did not reduce food intake or body weight in *Gfral*⁻ᐟ⁻ mice (Fig. 5B–D). Consistently, the beneficial effects of artesunate on glucose tolerance were abolished in *Gfral*⁻ᐟ⁻ mice (Fig. 5E, F). To further confirm whether the anti-obesity effect of artesunate is mediated by the GDF15/GFRAL axis, we quantified the percentage abundance of active c-Fos⁺ neurons within the AP, following a bolus intraperitoneal dose of artesunate. Similar to the results observed in mice with sh*GDF15*, artesunate-induced neuronal activation was abrogated in *Gfral*⁻ᐟ⁻ mice (Fig. S12A, B). These results suggested that artesunate-induced the production of GDF15, which in turn activated GFRAL⁺ neurons in mice. The GDF15/GFRAL axis is indispensable for the body weight-regulatory functions of artesunate.

### Artesunate induces weight loss without induction of nausea and malaise

Given the anorexic nature of GDF15, we examined whether the appetite-suppressing effect of artesunate is a consequence of emesis resulting from elevated levels of GDF15. Despite no significant changes in kaolin consumption (pica behaviour) in artesunate-treated rats (Fig. S13A), the body weight and food intake of rats with diet-induced obesity were considerably reduced in response to artesunate treatment (Fig. S13B, C). We then addressed whether artesunate may result in conditioned taste aversion, which usually occurs in response to exposure to toxic or aversive substances. Artesunate treatment did not lead to significant changes in saccharin preference or water consumption (Fig. S13D). These data suggested that the weight-reducing benefits conferred by artesunate treatment are unlikely related to nausea and malaise. To further verify the safety of artesunate for the treatment of obesity, we examined the clinical pathology parameters in normal and DIO mice treated with artesunate for 60 days using regular blood tests. Mice with long-term treatment with artesunate did not show any sign of clinical abnormalities compared to mice treated with vehicle controls (Table S2). Instead, they exhibited dramatic improvements in hepatic and metabolic parameters (Table S2). To verify the safety of artesunate in the neurological system, we conducted the Morris water task to assess the neurocognitive functions of artesunate-treated mice. During the test, total locomotor movement, escape latency, and swimming velocity were similar in vehicle- and artesunate-treated mice (Fig. S14A–D). Unlike GLP-1-based therapeutics, which are usually associated with gastrointestinal events, artesunate treatment did not result in overt changes in gastrointestinal transit and secretion in DIO mice (Fig. S15A–D). Considering the

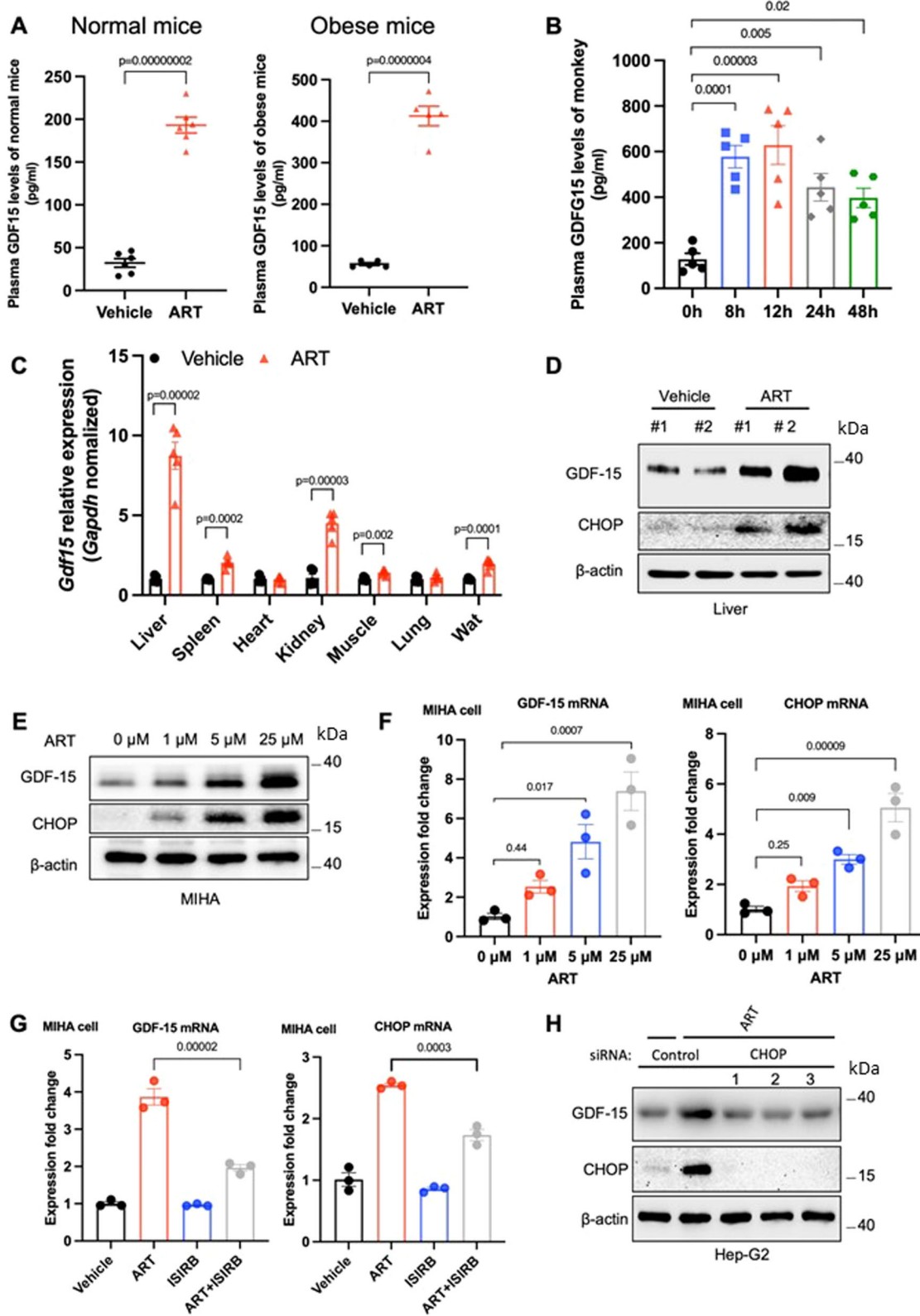

absence of malaise and other pathological changes in artesunate-treated mice and macaques, our findings suggested that artesunate controls body weight in a safe and efficacious manner.

## Discussion

Despite tremendous resources and efforts in obesity research over the years, medications with good safety and tolerability profiles for obesity remain limited. The discovery of GDF15, a stress-responsive hormone for the non-homeostatic control of body weight, has created hope for the treatment of obesity. However, the anorexic nature of GDF15 and its association with cancer cachexia and chemotherapy-induced anorexia have hampered its therapeutic potential in the treatment of obesity. Concentration is a key determining factor for the anorectic effect of GDF15. Low levels of GDF15 ($<200$ pgml$^{-1}$) are not sufficient to

**Fig. 3 | Artesunate increases GDF15 production through an integrated stress response pathway. A** Circulating GDF15 levels after a subcutaneous administration of artesunate (ART; 20 mg/kg) to lean, male wild-type mice (right) (*n* = 6 per group) and mice with diet-induced obesity (DIO) (left) (*n* = 5 per group). **B** serum GDF15 levels in spontaneously obese cynomolgus macaques in response to a single intravenous administration of ART (6.4 mg/kg) over 48 h (*n* = 5 per group). **C** qPCR analyses of Gdf15 expression in indicated organs of ART-treated mice with DIO (*n* = 5 per group). **D** Immunoblot analyses of CHOP and GDF15 expression in the livers derived from DIO mice treated with ART or vehicle. **E, F** Western blotting analyses (**E**) and qPCR analyses (**F**) of GDF15 and CHOP expression in MIHA cells upon 24 h-treatment of ART with different dosages (*n* = 3 independent experiments for the western blotting and qPCR analyses) **G** qPCR analyses of CHOP and GDF15 levels in MIHA cells cotreated with ART and ISIRB (*n* = 3 independent experiments for the qPCR analyses). **H** Immunoblot analysis of CHOP and GDF15 levels relative to β-actin in Hep-G2 cells with siRNA-mediated knockdown of CHOP in response to ART treatment (*n* = 3 independent experiments for western blotting). Data are reported as mean ± s.e.m; two-tailed unpaired Student's *t*-test (**A**; **C**) or one-way ANOVA followed by Tukey's test (**B**; **F**; **G**). Source data are provided as a Source Data file.

modulate appetite, whereas high levels of GDF15 (approximately 900 pgml$^{-1}$) have been shown to induce emesis or emesis-like behaviours in rodents. Here, we have identified artesunate as a moderate inducer of GDF15, at ~400 pgml$^{-1}$ in rodents and ~600 pgml$^{-1}$ in non-human primates. We further demonstrated that it prophylactically and therapeutically normalised body weight and alleviated metabolic dysfunction in a GDF15/GFRAL-dependent manner without the induction of nausea/malaise and any other pathological changes in both mice with high-fat diet-induced obesity and cynomolgus macaques with spontaneous obesity. Furthermore, we found that, in addition to the suppression of appetite, artesunate maintained the energy expenditure in obese mice. This observation is in line with the study showing that GDF15 encounters the compensatory reduction in energy expenditure associated with calorie restriction[40]. In addition to artesunate, other pharmacological GDF15 inducers, including camptothecin and metformin, have been shown to confer anti-obesity effects with comparable efficacies[41–43]. These results highlighted the potential of drugs targeting the GDF15/GFRAL axis for the treatment of obesity.

Artesunate, an artemisinin derivative, has been approved for the treatment of severe malaria by the FDA and other regulatory authorities for years. It is safe, well-tolerated, and can be administered to infants, children, and pregnant women with low toxicities[44]. The most common side effects of artesunate are nausea and dizziness, but these effects are mostly associated with acute malaria in many patients[34]. More serious complications associated with the use of artesunate, including neutropenia, anaemia, and haemolysis, have been found to be very rare, with an estimated risk of approximately 1 reaction per 3000 treatments[35]. Again, these effects are mostly attributed to severe malaria in patients rather than to drugs[35]. To date, there have been no comprehensive reports on the safety assessment of the long-term use of artesunate in malaria-free healthy subjects. Artesunate safety was examined in different animal species, such as rodents, rabbits, dogs, and macaques, and no adverse health effects were observed, even up to a dose of 100–150 mg/kg[45]. Artesunate modulated neurological activity at a dose of 128 mg/kg in rhesus macaques, but brain autopsy confirmed no pathological features. Thus, the tested doses of artesunate at 20 mg/kg for mouse and 6.4 mg/kg for monkey in this study are safe and well-tolerated for the medical management of obesity. In this study, we demonstrated that long-term use of artesunate reduces food intake, promotes weight loss, and ameliorates metabolic dysfunctions in both mice with high-fat diet-induced obesity and cynomolgus macaques with spontaneous obesity without the induction of nausea/malaise and any other toxicity effects. However, further clinical trials are needed to validate the effectiveness and safety of artesunate for the treatment of obesity in humans. Growing evidence has shown the anti-obesity potential of artemisinin and its derivatives[36]. Although the anti-obesity effect of artemisinin has been previously revealed in rodents, whether the observations obtained from mouse studies can be translated to higher species remains unknown, and the mechanism underlying the control of energy homeostasis remains unclear and controversial. There are conflicting results regarding the inducing effect of artemisinin on insulin production via the regulation of GABA-mediated conversion of alpha cells to functional beta-like cells[46,47]. Our present findings revealed that the body weight-lowering and insulin-

sensitising effects of artesunate are predominantly mediated by GDF15/GFRAL signalling, a key pathway for the non-homeostatic regulation of body weight. However, the mechanism underlying the appetite-regulatory functions of artesunate is not necessarily restricted to its inducing effects of GDF15 only. Further investigations are required to delineate the mechanism of action of artesunate, especially for its potential regulatory effects on GFRAL levels, a key factor determining the potency of GDF15/GFRAL signalling in the context of obesity[48]. Furthermore, we showed that artesunate controls body weight and food intake in both obese primates and non-primates without adverse effects. Compared with equimolar doses of other well-characterised anti-obesity drugs, including metformin (MEF) and liraglutide (LIR), treatment with artesunate showed superior potency in body weight regulation. This knowledge is of key importance given the current proposals to repurpose drugs to improve metabolic health in humans.

Our results indicated that artesunate normalises body weight both prophylactically and therapeutically, while reversing metabolic abnormalities commonly associated with obesity in a GDF15/GFRAL axis-dependent manner. Artesunate has a number of advantages over other GDF15 agonists, such as recombinant GDF15 proteins, in combating obesity. Unlike many anti-obesity drugs with unacceptable adverse effects, artesunate has been used as a potent anti-malarial drug with a good profile of safety and efficacy for decades, which reduces safety concerns in drug development. Indeed, we demonstrated that artesunate efficiently induces weight loss in both obese mice and macaques without inducing toxic responses and nausea/malaise. In non-obese animals fed a standard diet, artesunate administration had no significant effects, which is another guarantee of safety. In addition, artesunate, which can be generated using efficient and economically viable synthetic approaches, is much more cost-effective than recombinant proteins. Given the strong anti-obesity effects of artesunate in preclinical models, our results suggested that artesunate could be an effective therapeutic approach for the treatment of obesity, a global public health crisis with an urgent need for effective interventions. Further clinical trials are needed to validate the effectiveness and safety of artesunate for the treatment of obesity in humans.

Limitations: previous studies have demonstrated that artemisinin derivatives prevent obesity in rodent models by promoting energy expenditure by inducing browning in white adipose tissues via an unknown mechanism[36]. Our study suggests that reduced energy intake is a major contributor to the anti-obesity effect of artemisinin. It is possible that these two mechanisms may synergistically contribute to the weight-reducing function of artemisinin, which warrants further investigation.

## Methods
### Mouse
C57BL/6 J male wild-type (WT) mice and Sprague-Dawley rats were provided by The Chinese University of Hong Kong. *Gfral*$^{-/-}$ were purchased from Cyagen Biosciences Inc and housed at the Hong Kong Baptist University. *Gfral*$^{-/-}$ and their WT littermate controls were generated *on* C57BL/6 background as described previously[48]. All mice and rats were housed in a temperature-controlled room with 12-h light/12-h dark cycle with ad libitum access to food and water. Only male mice

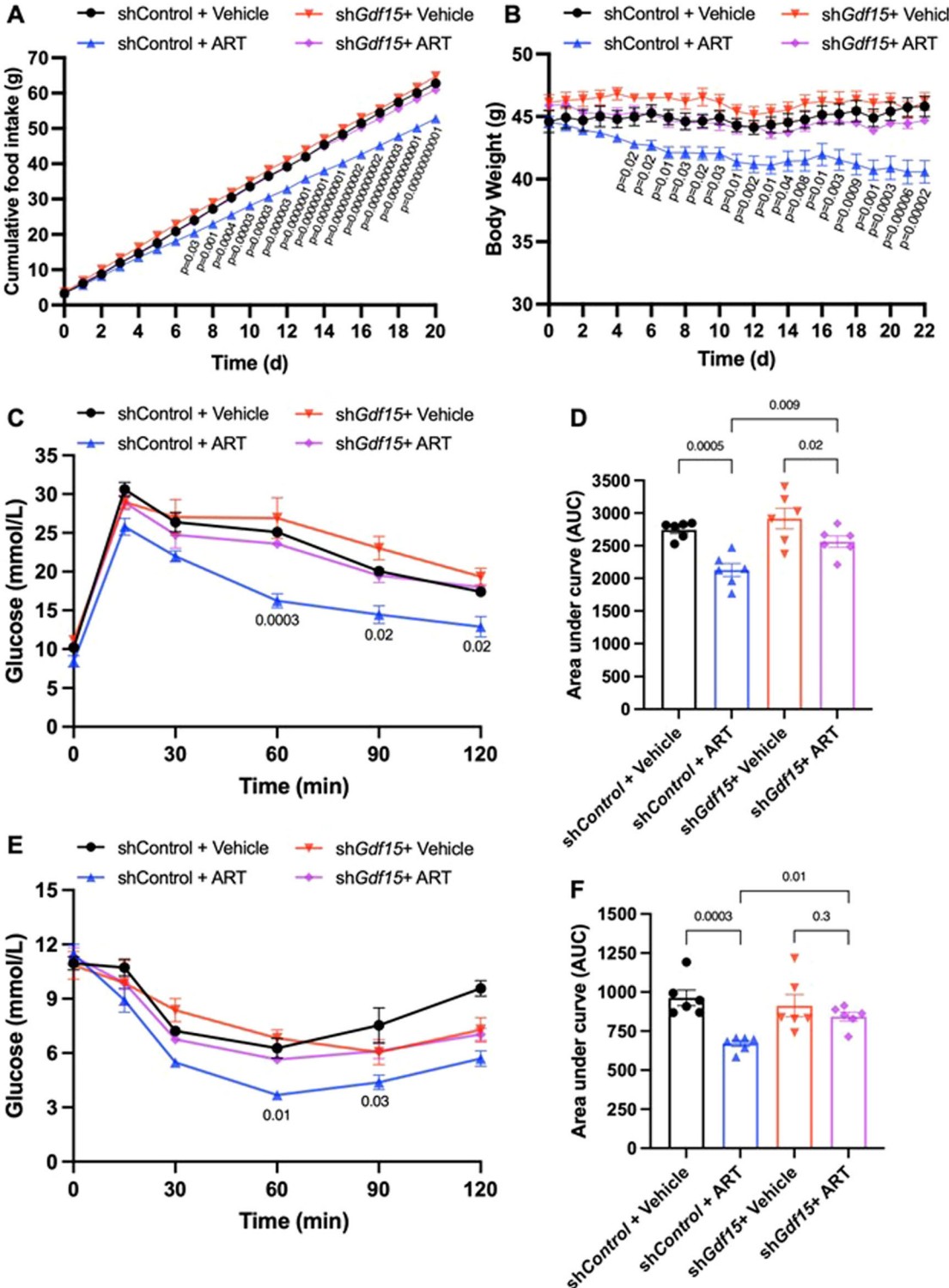

**Fig. 4 | Artesunate-mediated anti-obesity effect is GDF15 dependent.** Diet-induced obese mice injected with adeno-associated virus expressing shControl or shGDF15 were treated daily with either vehicle or artesunate (ART; 20 mg/kg) for 21 days. **A** Cumulative food intake and **B** changes in body weight in obese shGDF15 mice with ART or vehicle over the treatment period ($n = 6$ for all treatment groups) **C**, **D** Plasma glucose level and area under the curve (AUC) during the intraperitoneal glucose tolerance tests ($n = 6$ for all treatment groups) **E**, **F** Plasma glucose level and AUC during the insulin tolerance tests ($n = 6$ for all treatment groups). Data are reported as mean ± s.e.m.; two-way ANOVA followed by Tukey's multiple comparison post-hoc test (**A**–**C**) and Fisher's LSD post-hoc test (**E**), one-way ANOVA followed by Fisher's LSD post-hoc test (**D**, **F**). For the curve plots, the $p$-value represents the comparison between shControl + ART and shGDF15 + ART treatment groups (**C**, **E**). Source data are provided as a Source Data file.

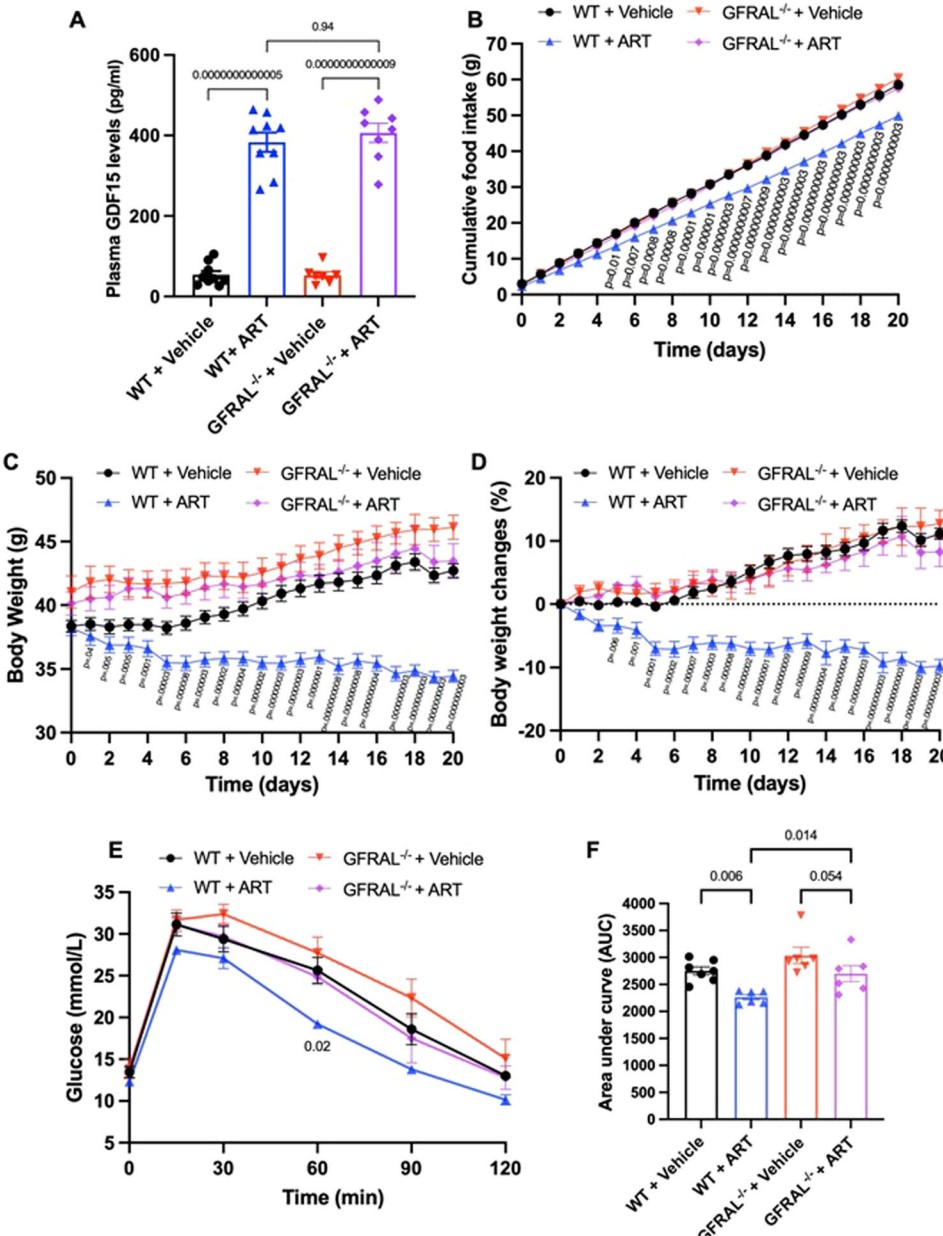

**Fig. 5 | GDF15/GFRAL axis is necessary for artesunate-induced food intake reduction and weight loss. A** Serum GDF15 levels 24 h upon a single intraperitoneal administration of vehicle or artesunate *(ART; 20 mg/kg)* in wild-type (WT) and *Gfral−/−* mice (*n* = 9 per group). **B–D** Changes in cumulative food intake (**B**) and body weight (**C**, **D**) in WT and *Gfral−/−* mice in response to intraperitoneal administration of vehicle or ART (20 mg/kg) over 21 days (*n* = 6 per group). **E, F** Intraperitoneal glucose tolerance test (**E**) and area under the comparison (AUC)

(**F**) after 21 days of treatment with vehicle or ART (20 mg/kg) in WT and *Gfral−/−* mice (*n* = 6–7 per group). Data are presented as mean ± SEM; one-way ANOVA followed by Šídák multiple comparison test (**A**) & Fisher's LSD post-hoc test (**F**), two-way ANOVA followed by Tukey's multiple comparison post-hoc test (**B–E**). For the curve plots, the *p*-value represents the comparison between WT + ART and GFRAL−/− + ART treatment groups (**B–E**). Source data are provided as a Source Data file.

and rats were used in the experiments. All experimental protocols involving mice and rats were reviewed and approved by the Use of Human & Animal Subjects in Teaching & Research (HASC) at Hong Kong Baptist University and in compliance with the Department of Health, Hong Kong.

### High-fat-diet feeding
One week after weaning, mice were randomly selected and assigned to one of two dietary treatments: control diet (Research diets #D12450J) or HFD (Research diets #D12492). On a caloric basis, the control diet is with 10 kcal% fat whereas HFD is with 60 kcal% fat.

### Drug treatment
For ART treatment, mice received 5 mg/kg or 20 mg/kg of ART (MB7317, Meilunbio) or vector (5% sodium bicarbonate mixed with 0.9% physiological saline) intraperitoneally (i.p.) according to the different experimental requirements until the end of the experiment, and the inoculation period is indicated in the different mouse experiments and in each figure legend.

For metformin-treated mice, Metformin hydrochloride (M107827, Aladdin) was dissolved in ddH$_2$O, and intragastric administration (i.g.) with Metformin hydrochloride at a concentration of 200 mg/kg according to the body weight of the mice.

For liraglutide treatment, liraglutide (L276430, Aladdin) was dissolved in 0.9% saline, and mice were injected subcutaneously (s.c.) at a concentration of 100 ug/kg of liraglutide according to body weight.

## Animal protocol 1: effect of ART on wild-type mice in HFD challenge

C57BL/6 J mice at the age of 8 weeks were randomly assigned for the experiment. During HFD challenge, mice were treated with intraperitoneal injections of vehicle control and ART every two days (vehicle, ART low-dose treatment group 5 mg/kg, ART high-dose treatment group 20 mg/kg, 8 mice in each group). The body weight and food intake of the mice were monitored daily. At the indicated time points, mice were anaesthetised in an isoflurane-saturated chamber after 6 h of starvation and then sacrificed by cardiac puncture.

## Animal protocol 2: effect of ART on mice with diet-induced obesity

C57BL/6 J mice were fed HFD for 16 weeks until their mean body weight was 43.35 g ± 1 g. Mice were then randomly assigned to the experiment (vehicle, ART treatment group, 6 mice per group). Body weight and food intake were monitored daily. At the indicated time points, mice were anaesthetized in an isoflurane-saturated chamber after 6 h of starvation and then sacrificed by cardiac puncture.

## Animal protocol 3: pair feeding

Pair-fed vehicle-treated animals received the amount of food equivalent to that of the corresponding ART-treated group. Body weight and food intake were measured daily.

## Animal protocol 4: lean mice treated with ART

C57BL/6 J mice (lean, weighing 23 to 25 g) were injected intraperitoneally with the vehicle or ART for 2 weeks. Body weight and food intake were monitored daily.

## Animal protocol 5: effect of ART in *GDF15 knockdown mice*

C57BL/6 J mice were fed with high-fat diet for 16 weeks until their mean body weight was 45.59 g ± 0.64 g. AAV8-shControl and AAV8-sh*GDF15* virus produced by the pAAV-U6-shRNA (Gdf15)-CBh-EGFP-WPRE vector plasmid were purchased from Obio Technology Ltd. This vector plasmid was an AAV vector of serotype 8 under the control of U6 promoter, a well-documented AAV vector for specific transgene expression in hepatocytes. They were delivered by tail vein injection ($1 \times 10^{11}$ copies per mouse). Mice were studied 2 weeks after AAV8 infection. Mice were then randomly assigned to the experiment (vehicle, ART treatment group, 6 mice per group) groups and injected accordingly with vehicle or ART every two days for 3 weeks. Mice were monitored daily for body weight and food intake. Mice were anaesthetized in an isoflurane-saturated chamber and then sacrificed by cardiac puncture for tissue harvest.

## Animal protocol 5: effect of ART on *Gfral⁻/⁻* mice

Experimental cohorts of *Gfral⁻/⁻* and *Gfral⁺/⁺* mice were obtained from het x het breeding pairs[48]. *Gfral⁺/⁺* and *Gfral⁻/⁻* mice at 12 weeks of age were treated with HFD for 4 weeks, and each mouse was intraperitoneally injected with vehicle or ART every two days for 3 weeks, and their body weight and food intake were monitored daily. On day 21, mice were euthanized by terminal anaesthesia 6 h after treatment, and plasma was obtained. Tissues were freshly frozen on dry ice and stored at −80 °C.

## Animal protocol 6: cynomolgus monkey efficacy study

For the non-human primate model of obesity, eight male macaques with a mean age of 12.5 years, MI > 40 and body condition scoring >4.5[49] were selected for training, which focused on simulated blood collection for a 2-week training period, after which five macaques were selected for formal experiments based on the training performance and basic biochemical data. Only male macaques were used for the experiments. Macaques were treated with intravenous injections of vehicle every two days through 2 weeks. After a 2-week washout period, macaques were treated with ART (6.4 mg/kg) every two days intravenously for 2 weeks. Food intake was counted daily, and animal body weight was measured every 3 days. These macaques were housed individually in stainless steel cages and maintained at ambient temperature (16–26 °C) and humidity (~60%) under a 12-h (h) light/dark cycle. Breakfast (80 g) and dinner (120 g) were the standard maintenance diet for non-human primates (Jiangsu Synergy Pharmaceutical and Biological Engineering Co., Ltd.), and lunch was apple (70 g). Animals were fed water ad libitum through a waterer. All husbandry conditions and procedures were approved by Guangzhou Huazhen Bioscience Co., Ltd (IACUC) and conformed to the ethical guidelines of their Institutional Animal Care and Use Committee. The husbandry facilities were accredited by the American Association for Assessment and Accreditation of Laboratory Animal Care (AAALAC).

## Animal protocol 7: effect of ART on kaolin consumption in HFD-induced obese rats

The kaolin intake experiment is a well-recognised model for assessing pica behaviours in Sprague-Dawley rats[50]. 8-week-old male rats were fed with HFD to a body weight of 560 g–610 g. One week prior to the start of treatment, kaolin was added to allow the animals to acclimatize. The rats were then divided into 3 weight-matched groups (5 rats per group) and injected intraperitoneally daily with vehicle, ART (14 mg/kg). Food, kaolin intake, and body weight were measured daily.

## Animal protocol 8: comparison of ART with liraglutide and metformin for the treatment of obese mice

Eight-week-old male mice were treated with HFD for 8 weeks to a mean body weight of 44.5 g (±2.1), after which the mice were equally assigned by body weight to Vehicle (44.4 ± 0.7 g), LTR (44.4 ± 1.2 g), MEF (44.4 ± 0.8 g), and ART (44.6 ± 0.7 g) group. DIO mice were treated with Vehicle, LTR, MEF, and ART every two days, and body weight and food intake were measured daily, followed by GTT and ITT assays on DIO mice. Mouse serum was collected for biochemical analysis. Thereafter, mice were anaesthetized in an isoflurane-saturated chamber and then sacrificed by cardiac puncture for tissue harvest.

## Animal protocol 9: effect of ART on gastrointestinal parameters

C57BL/6 J mice were treated with vehicle or ART for 8 weeks and monitored daily for body weight. To measure the gastrointestinal transit time, individually caged mice were orally gavaged with 150 μl of Carmine red solution prepared as 6% (w/v) solution in 0.5% methylcellulose. Time taken to release the first carmine red faecal pellet after the initial gavage was recorded as total intestinal transit time. Faecal pellet output was calculated by counting the number of pellets released from each mouse in an observation period of 1 h. Collected faecal pellets were dried overnight in an oven and measured for the dry weight. Percentage difference between the initial faecal wet weight and dry weight indicated faecal water content.

## Glucose tolerance test and Insulin tolerance test

To perform GTT, mice were fasted overnight and received IP injection of 20% glucose (2 g kg-1 of body weight, Sigma–Aldrich). For ITT, mice fasted for 6 h received IP injections of insulin (0.5 units kg-1 body weight, Sigma–Aldrich). Tail glucose levels were measured at the indicated times (0, 30, 60, 90 and 120 min) after injection.

## Antibodies

The antibodies used in this study include the following: anti-GFRAL antibody (ab214929, Abcam, 1:100 dilution for immunofluorescent staining); anti-FOS (FOS, 2250, Cell Signaling Technology, 1:200 dilution immunofluorescent staining); anti-GDF15 (27455-1-AP, Proteintech, 1:2,000 dilution for western blot); anti-CHOP (15204-1-AP, Proteintech, 1:2,000 dilution for western blot); anti-β-actin (sc-47778, Santa Cruz, 1:2,000 dilution for western blot); m-IgGκ BP-HRP (sc-516102, Santa Cruz, 1:5,000 dilution); goat anti-rabbit antibody conjugated with HRP (sc-2357, Santa Cruz, 1:5,000 dilution); Alexa Fluor 488-conjugated donkey anti-sheep antibody (A-11015, Invitrogen, 1:500 dilution); Alexa Fluor 594-conjugated goat anti-rabbit (A11012, Invitrogen, 1:500 dilution).

## Biochemical analysis

GDF15 levels in mouse plasma were measured by enzyme-linked immunosorbent assay (ELISA, R&D Systems, MGD150). GDF15 in macaque plasma was measured by enzyme-linked immunosorbent assay (ELISA, Cloud-clone, SEC034Si). Plasma insulin was assessed after 12 h of fasting using the Cloud-clone Rat/Mouse Insulin ELISA Kit (CEA448Mu) according to the manufacturer's instructions.

## Cell treatment

Hep-G2 cells were purchased from ATCC (HB-8065). MIHA cell line was a gift from the University of Hong Kong, which was originally purchased from the National Collection of Authenticated Cell Cultures (https://www.cellbank.org.cn/); the sex origins of cell lines are unknown. MEF cells were isolated using 13.5 days developed embryos, digested into single-cell suspensions using 0.25% trypsin-EDTA and cultured for 2 generations for experiments. Hep-G2, MIHA and MEF cells were cultured in DMEM (Gibco) supplemented with 10% FBS and penicillin/streptomycin (100 ng/ml). Cells used in the experiments were tested free of mycoplasma contamination. All cell lines were passaged for no more than 10 generations. For siRNA transfection, 80–90% confluent cells were transfected using Lipofectamine RNAi-MAX (Invitrogen). Two technical replicates and three biological replicates were used for each cell experiment.

## Western blotting

Western blot was performed as previously described:[51] total protein was extracted from cell or tissue samples using RIPA buffer composed of 25 mM Tris-HCl, 150 mM NaCl, 1% NP-40, 0.1% SDS, 1% sodium deoxycholate and complete protease cocktail (Roche) on ice. Samples were sonicated four times for 5 s on ice water at 15% power (Ningbo Scientific Biotechnology). Tissue lysates were centrifuged at 4 °C for 10 min at 14,000 $g$ and supernatant was mixed with 4× LDS Sample Buffer (NP0007) and boiled for 10 min at 70 °C. The 10-μg protein samples were separated by 4–12% SDS–PAGE with MOPS-SDS running buffer and transferred to polyvinylidene difluoride membranes (Bio-Rad Laboratories). Membranes were blocked with 5 QuickBlock™ Western Blocking Solution (P0252, Beyotime Biotechnology) for 1 h and then incubated with primary antibodies prepared using Western Primary Antibody Diluent (P0256, Beyotime Biotechnology) at 4 °C overnight. Membranes were incubated with HRP-conjugated secondary antibodies prepared using QuickBlock™ Western Secondary Antibody Diluent (P0258, Beyotime Biotechnology), and incubated for 1 h at room temperature. Immunoreactions were captured by X-ray films detecting chemiluminescence using an ECL kit (GE Healthcare). The full scan blots can be found in the Fig. S16.

## Biochemical analysis of serum samples

Whole blood specimens are placed at room temperature for 2 h and then centrifuged at 24 °C for 15 min at 3000 rpm, and the supernatant is taken for immediate detection, or the specimens are divided and stored at −80 °C. ALT, AST, TG, TC, HDL, and LDL are measured according to the kit manufacturer's instructions (Shenzhen Rayto Life Sciences) and are automatically determined using a fully automated biochemistry instrument (Shenzhen Rayto Life Sciences, Chemray 800).

## Real-time qPCR

Total RNA was extracted from cells or tissues using the RNAeasy™ Animal RNA Isolation Kit with Spin Column Reagent (Beyotime) according to the manufacturer's protocol. A total of 500 ng RNA was reverse-transcribed using PrimeScript RT master mix (Takara Bio). The resulting cDNA templates amplified with specific primers of target genes were analysed by the ABI ViiA 7 Real-time PCR System (Applied Biosystems) using real-time PCR kits (Takara Bio). The sequences of primers used include *Gdf15*: AGCCGAGAGGACTCGAACTCAG (forward); GGTTGACGCGGAGTAGCAGCT (reverse); *Gapdh*: GAGCCA AAAGGGTCATC (forward); GTGGTCATGAGTCCTTC (reverse); GDF15: CAACCAGAGCTGGGAAGATTCG (forward); CCCGAGAGATACGCAG GTGCA (reverse); CHOP: GGTATGAGGACCTGCAAGAGGT (forward); CTTGTGACCTCTGCTGGTTCTG (reverse); beta-actin: CACCATTGGCA ATGAGCGGTTC (forward); AGGTCTTTGCGGATGTCCACGT (reverse).

## Hematoxylin and Eosin (H&E) staining

Freshly extracted tissues were fixed overnight in 4% PFA and subsequently dehydrated with gradient alcohol using a gradient dehydrator (LTP-200, HASCO), followed by wax immersion of the samples. The wax-impregnated tissues were embedded in an embedding machine (ES-500, Huaspeed Technology), and the trimmed wax blocks were subsequently sectioned on a paraffin microtome. For HE staining, paraffin sections were dewaxed to water and subsequently stained with hematoxylin for nuclei and eosin for cytoplasm, and images were acquired and analysed using a scanner (3D histech, Pannoramic MIDI).

## Oil Red O staining

The collected fresh tissues were frozen sectioned using OCT embedding and subsequently stained using Oil Red O Staining Kit (C0157S, Beyotime). Specifically, an appropriate amount of staining washing solution was added to cover the samples for 20 s, followed by aspiration of the staining washing solution. The prepared Oil Red O staining working solution was added dropwise to the sections for staining for 10–20 min, removal of Oil Red O staining working solution was added dropwise to the sections and left for 30 s, then the staining washing solution was removed, and the sections were immersed in distilled water and washed on a shaker for 20 s, and the nuclei were re-stained using hematoxylin staining solution (C0107), sealed and then observed and photographed under the microscope.

## Immunohistochemistry

For all immunohistochemistry experiments, mice were anaesthetized using isoflurane and perfused with ice-cold saline (0.9%) followed by 4% paraformaldehyde (PFA) in 0.1 M PBS. Brains were post-fixed overnight in 4% PFA and preserved with 30% sucrose (w/v) at 4 °C for 48 h. 10 μm serial coronal sections were prepared using Cryostat (Leica Biosystems). Coronal sections within the distance of −7.48 to −7.64 mm from bregma were checked for area postrema region by observing through the microscope for the anatomical landmarks as described in the mouse brain atlas[52]. Sections were permeabilized with 0.1% Triton-X 100 in 0.1 m PBS for 15 min and blocked with 1% BSA in PBST for 1 h. Sections were incubated overnight with primary antibodies at 4 °C, followed by secondary antibodies at room temperature for 1 h. Slides were mounted using Prolong Gold with or without DAPI (Thermo Fisher). Images were obtained using a Confocal Laser Scanning Microscope (Leica TCS SP8). Gfral⁺ cFos⁺ cells were counted in

every third serial section and at least 5 sections were quantified to span the entire AP region.

## Energy expenditure

Mice were profiled for energy expenditure (EE) in Promethion metabolic cages (Sable Systems International, Las Vegas, NV). Mice were singly housed in Promethion cage equipped with a food hopper, and water spigot connected to a load cell for continuous real-time monitoring of food (g) and water intake (g). Mice were acclimatised for one week before the experiment start-date to minimise the stress of the new environment. After the acclimatisation, body composition and weight were recorded for the baseline values. X/Y/Z beam-break systems were positioned around the cage to detect the mouse activity and to facilitate the quantitative measurement of physical activity. Cages were connected to an oxygen sensor and carbon dioxide sensor to examine the rate of oxygen consumption ($VO_2$) and rate of carbon dioxide emission ($VCO_2$) in ml/min. Respiratory exchange ratio (RER) was calculated as the ratio of $CO_2$ production to $O_2$ consumption. Energy expenditure (EE) (per kg body weight) was calculated using the Weir equation[53]. Analysis of covariance (ANCOVA) was performed with body weight as a covariate for EE measurements. All the raw data were processed using a macro interpreter provided by Sable Systems and data acquisition was coordinated by MetaScreen software (v.2.3.15.12).

## Neurobehavior assessment

Morris water maze was performed as described previously[54]. Mice were examined for six days (2 trials/day) to navigate and locate the hidden platform present in a circular pool filled with opaque water. Visualisation cues were positioned around the water maze and starting position was selected randomly every day. Mouse movement were tracked automatically using the Any-Maze video tracking recording system. Each mouse was analysed for escape latency (time taken to locate the hidden platform), path length (distance travelled to reach the hidden platform) and average swimming speed.

## Statistical analyses

Data was expressed as means ± standard error (mean ± SEM) and the statistical significance was set at $p < 0.05$. The statistical analyses were performed with GraphPad Prism 8.0 (GraphPad Software, LLC, San Diego, CA) and tested by either Student's $t$-test, one-way or two-way ANOVA as indicated in the figure legends. The sample size and number of replicates for each experiment are described in the figure legends. In vivo experiment involving a minimum of three independent age-matched mice and randomisation were performed where applicable. The sample size was estimated with the power of the statistical test performed.

## Reporting summary

Further information on research design is available in the Nature Portfolio Reporting Summary linked to this article.

# Data availability

All data supporting the findings of this study are available within the paper and its Supplementary Information. Source data are provided with this paper. Any additional information is available upon request to the corresponding author (Hoi Leong Xavier Wong, xavierwong@hkbu.edu.hk). Source data are provided with this paper.

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

## Acknowledgements

The presented work was kindly supported by the General Research Fund (12102020 and 22104123) (Wong HLX), Health and Medical Research Fund (08793626) (Wong HLX), Excellent Young Scientist Fund by National Natural Science Foundation of China (32322091) (Wong HLX) and Innovation and Technology Fund (ITS/058/22MS) (Wong HLX).

## Author contributions

X.G. performed most of the experiments. C.K.W., G.S., H.J., P.A., W.J., Z.Y., and Z.L. helped to collect samples and performed some of the experiments. M.A.K., S.M., and U.M. provided experimental materials for experiments. K.H.Y., L.A., C.K.M., and X.P. contributed to the discussion. X.G., P.A., and H.L.X.W. designed the experiments and prepared the manuscript. H.L.X.W. and Z.X.B. obtained research funding and supervised the project.

## Competing interests

The authors declare no competing interests.

## Additional information

[1]School of Chinese Medicine, Hong Kong Baptist University, Hong Kong SAR, China. [2]Centre for Chinese Herbal Medicine Drug Development Limited, Hong Kong Baptist University, Hong Kong SAR, China. [3]Fujian Provincial Key Laboratory of Innovative Drug Target Research and State Key Laboratory of Cellular Stress Biology, School of Pharmaceutical Sciences, Xiamen University, Xiamen, China. [4]Institute of Biotechnology-HILIFE, University of Helsinki, Helsinki, Finland. [5]Icosagen Ltd., 61713 Tartu, Estonia. [6]Department of Biomedical Sciences, City University of Hong Kong, Hong Kong SAR, China. [7]Department of Neurology, the First Affiliated Hospital of Guangzhou Medical University, Guangzhou, China. [8]These authors contributed equally: Xuanming Guo, Pallavi Asthana. ✉e-mail: pallavi@hkbu.edu.hk; bianzxiang@gmail.com; xavierwong@hkbu.edu.hk

