## [Peer Review File · Nature Communications]

Reviewers' comments:

Reviewer #1 (Remarks to the Author):

In the manuscript "Artesunate treats obesity in non-human primates through GDF15/GFRAL signaling axis" by Guo and colleagues, the authors demonstrate the effectiveness of the anti-malaria drug Artesunate in preventing and treating obesity in mice and non-human primates. The study shows that Artesunate prevents weight gain in mice on a high-fat diet and leads to weight loss in obese mice. In addition, the authors observe that Artesunate treatment causes weight loss in obese primates in a sub-chronic treatment regimen. The study also demonstrates that Artesunate improves glucose homeostasis and insulin sensitivity in obese mice. While the weight-lowering effects of Artesunate in obese mice have been published previously, the novelty of this study lies in the treatment of non-human primates and the connection to GDF15/Gfral.

It would strengthen the paper, if a study in which Artesunate is administered to GDF15 knock-out mice would be performed. Alternatively, the authors could treat wild-type obese mice with Artesunate and a GDF15 neutralizing antibody to induce weight gain, which would support their hypothesis of the role of GDF15-GFRAL signaling.

In addition, it would be crucial to demonstrate that the antibody for GDF15 is specific. Our lab has tried pretty much every commercially available AB. All of them show a band in GDF15 KO tissue, demonstrating that they are not specific.

The authors only present data on food intake for short periods of time, and it would be helpful to see data for the entire treatment period. The authors claim that Artesunate "restores" lipid metabolism, but this claim should be supported by data from age-matched mice on a low-fat diet.

The authors also need to provide more information on the number of biological and technical replicates in the cell experiments and the passage number of the cells. In addition, it would be helpful if the authors could address the adipose tissue inflammation seen in Artesunate-treated mice and provide data on energy expenditure and adipose tissue metabolism in the rodent studies to further explore the mechanism of action of Artesunate.

Overall, the manuscript lacks attention to detail and could benefit from more thorough proofreading and editing.

In conclusion, it would be beneficial if the authors could address the potential safety concerns associated with recommending Artesunate as a treatment for obesity. It is well known that Artesunate can cause severe side effects, such as renal failure, gastrointestinal issues, insomnia, neurological impairments, and fatigue, in a significant number of patients. Given the popularity of social media platforms for sharing diet and health advice, it is particularly important to carefully consider the potential risks and benefits of recommending Artesunate for obesity treatment. It would be in the interests of the authors and the journal Nature Communications to carefully communicate the results of this study and the potential risks associated with using Artesunate. It is important to note that Artesunate is not a safe medication for humans.

Reviewer #2 (Remarks to the Author):

These authors seek to make the case that, based on pre-clinical data, artesunate is a candidate drug to treat obesity, these actions being mediated through induction of GDF15. We are told that artesunate is a drug licenced in the USA for treatment of malaria. The authors demonstrate that artesunate raises GDF15 levels in mice, from approximately 50 to 200 pg/ml at the end of a 12 day period of daily intraperitoneal administration. This rise in GDF15 levels is mediated by induction of the stress response with most induction occurring in liver. These findings and serum levels are comparable to those changes reported to be induced by metformin treatment in mice. The artesunate has no impact on body weight of lean mice, presumably because of the very small rise in serum levels. It does induce significant weight loss in obese mice and this is accompanied by reduction in food intake and improvement in glucose homeostasis and liver steatosis. The authors have used germ line deleted GDF15 receptor knock out mice and shown that the effects of artesunate on body weight do not occur in these GFRAL knock out obese mice. At this dose of GDF15 they could identify no sickness behaviour based on pica behaviour, but they have not used a more sensitive test in rodents such as condition taste diversion. Artesunate has comparable effects in spontaneously obese primates in whom there is also no emesis reported.

The actions of artesunate on weight loss are similar to those of metformin and liraglutide in the doses used within the experimental protocol. No information is provided on serum GDF15 levels in either the metformin or the liraglutide treated animals.

The authors believe that this effect is largely mediated through actions on the mouse hind brain. Whilst this is probably true, the evidence they present for this is very sketchy and I would recommend that this part of the manuscript either be removed or substantially revised. If it is to be revised, photographs need to adequately display the whole or at least most of relevant hind brain sections, which require a much lower power view. The sections need to identify the anatomical regions in question and the exact regions in the brain that have been sampled, need to be identified. Highly selected powered views are not very useful in this instance, although they can be used as a supplementary source of information.

The authors have not sampled energy expenditure and physical activity in the treated rodents and this is a significant limitation. Based on the existing literature, the results of these studies are not likely to have a major impact on the outcome.

This manuscript would be improved by improving the evidence that the effects of artesunate were GDF15 mediated. This would involve treating mice with recombinant GDF15 to raise serum levels to that reported within the manuscript and observe if similar weight loss results. This could be done by giving GDF15 via an infusion pump, which would mimic a much smoother changes in serum level seen with induction of GDF15 secretion by artesunate.

Overall, the authors make a convincing case that artesunate is another in the growing list of compounds capable of inducing GDF-15 expression and raising its serum levels. It is not completely clear if this does or does not induce sickness behaviour in mice and this finding would be more convincing if they used condition taste aversion. However, this is a difficult area to evaluate in rodents and if it truly does fail to induce sickness behaviour, this is likely to be due to the modest rises in serum levels associated with the use of this compound or the slower rate of change in serum levels. It is possible that if used in humans to treat obesity, it might cause modest weight reduction, as is seen with metformin. Whether to determine if it has less gastrointestinal side effects in metformin, would require a human clinical trial to evaluate.

The referencing and background could do with some improvement. The primary references for GDF15's effect on body weight and obesity should be cited rather than random selected references. The issue of exercise, GDF15 and weight loss is controversial, with various publications showing differing results. Simply saying that induction of GDF15 in response to exercise is not sufficient to modulate food intake is misleading. This should either be discussed more fully or the sentence discussing this removed.

Reviewer #3 (Remarks to the Author):

In this manuscript the Authors describe the effect of artesunate in for the treatment of obesity in in non-human primates. Non human primates are important models for the understanding and treatment of obesity and related metabolic conditions including type 2 diabetes mellitus, the insulin-resistance metabolic syndrome and arterial hypertension. Previous extensive work by Folli F and collaborators have established mechanisms of insulin resistance and beta-cell failure in baboons.

The experiments in mice and non-human primates are well performed, and they point to GDF15 as an important mediator of satiety. Previous extensive work by the groups of Martin Myers jr and Jens Bruning have provided major insights into the neuronal populations which are involved in the mechanisms regulating satiety and appetite.

What is lacking in the present work is how this GDF15 mediated pathways interacts with other neuronal subpopulations to regulate satiety. One way to start to uncover this could be performing triple immunostainings with leptin, MCH and other hormones with GDF15.

To add power to the experiments in monkeys they should be performed with a cross-over design

Reviewer #1

In the manuscript "Artesunate treats obesity in non-human primates through GDF15/GFRAL signaling axis" by Guo and colleagues, the authors demonstrate the effectiveness of the anti-malaria drug Artesunate in preventing and treating obesity in mice and non-human primates. The study shows that Artesunate prevents weight gain in mice on a high-fat diet and leads to weight loss in obese mice. In addition, the authors observe that Artesunate treatment causes weight loss in obese primates in a sub-chronic treatment regimen. The study also demonstrates that Artesunate improves glucose homeostasis and insulin sensitivity in obese mice. While the weight-lowering effects of Artesunate in obese mice have been published previously, the novelty of this study lies in the treatment of non-human primates and the connection to GDF15/Gfral.

1. It would strengthen the paper, if a study in which Artesunate is administered to GDF15 knock-out mice would be performed. Alternatively, the authors could treat wild-type obese mice with Artesunate and a GDF15 neutralizing antibody to induce weight gain, which would support their hypothesis of the role of GDF15-GFRAL signaling.

Response: Thank you very much for the valuable comment. This is a good suggestion and therefore we examined the effect of artesunate on mice with AAV-mediated *Gdf15* knockdown. High fat diet-induced obese mice were injected with AAV expressing *Gdf15*-specific shRNA (*shGDF15*) to knock down GDF15 specifically in the liver in which artesunate-induced GDF15 is majorly derived from. We found that knockdown of hepatic GDF15 dramatically reduced the artesunate-induced elevation of GDF15 levels and abrogated the anti-obesity effect of artesunate in obese mice (**Figure 4**), which was in alignment with the data showing that GFRAL is essential for the bodyweight-regulatory effect of artesunate (**Figure 5**). These results demonstrate the key role of GDF15-GFRAL signaling axis in artesunate actions.

2. In addition, it would be crucial to demonstrate that the antibody for GDF15 is specific. Our lab has tried pretty much every commercially available AB. All of them show a band in GDF15 KO tissue, demonstrating that they are not specific.

Response: We used the GDF15 primary antibody from Proteintech (Cat No. 27455-1-AP), which has been referred by numerous peer-reviewed papers (Bernasocchi, El Tekle et al. 2021, He, Shi et al. 2022) and validated in both GDF15 knockout and GDF15-overexpressed cell lines (Myojin, Hikita et al. 2021). To further validate the specificity of antibody, we examined the expression of GDF15 in the liver tissues derived from mice with AAV-*shGDF15*. We detected a specific band at 34kDa for GDF15 in wild-type livers and this band was greatly reduced in the livers with AAV-*shGDF15* (figure below).

3. The authors only present data on food intake for short periods of time, and it would be helpful to see data for the entire treatment period. The authors claim that Artesunate "restores" lipid metabolism, but this claim should be supported by data from age-matched mice on a low-fat diet.

***Response:** Thank you very much for the insightful suggestion. In our validation study, we confirmed that the artesunate-induced weight loss is mediated through GDF15/GFRAL axis and showed the cumulative food intake for the entire experimental period (Figure 4a, 5b). However, for the long-term effect of artesunate on food intake in HFD mice, we focussed on a specific time window of 8 days at the end of the experiment because of the following reasons – Firstly, metabolic cages are required to precisely calculate the food intake. Due to the limited availability of the instrument in our laboratory, it was unfortunately infeasible to perform the experiment in the metabolic cages for the entire treatment period (60 days). We hope that this technical limitation does not affect the reviewer’s confidence in our results. Secondly, the changes in bodyweight are a gradual process and the direct indicator of energy homeostasis. We found that artesunate induces the reduction in food intake without the disturbance of energy expenditure (Supplementary figure 9B-C). These results suggest that the weight-losing effect of artesunate is mainly attributed to the suppression of appetite.*

In addition, we examined the cholesterol levels in the serum samples from artesunate-treated high fat diet mice and aged matched mice on a low-fat diet. We found the levels of different cholesterol were reduced in obese mice by artesunate (Supplementary table 2). We therefore have revised the statement to “Artesunate treatment reduces adiposity and cholesterol levels”, which is more appropriate for the result description (Supplementary figure 5D-G).

4. The authors also need to provide more information on the number of biological and technical replicates in the cell experiments and the passage number of the cells. In addition, it would be helpful if the authors could address the adipose tissue inflammation seen in Artesunate-treated mice and provide data on energy expenditure and adipose tissue metabolism in the rodent studies to further explore the mechanism of action of Artesunate.

***Response:** Thanks for your advice. We have updated the experimental details in the method section (cell treatment). As advised, we performed the experiment to examine the effect of Artesunate on energy expenditure changes in high fat diet mice. We found that artesunate induces reductions in food intake without the disturbance of energy expenditure (Supplementary figure 9B-C). Furthermore, we examined the expression of genes associated with lipid metabolism (Lipoprotein lipase, *Lpl*; Acetyl-CoA Carboxylase Alpha, *Acaca*; Acetyl-CoA Carboxylase Beta, *Acacab*; Fatty acid synthase, *Fasn*; ATP-citrate lyase, *Acly*; 3-hydroxy-3-methylglutaryl-CoA reductase, *Hmgcr*; 3-hydroxy-3-methylglutaryl-CoA synthase 2, *Hmgcs2*; Stearoyl CoA desaturase 1, *Scd1*; Glycerol kinase, *Gk*) in liver and found that artesunate treatment did not alter their expression in the liver of obese mice (result below).*

5. Overall, the manuscript lacks attention to detail and could benefit from more thorough proofreading and editing.

Response: We used Elsevier Language Editing Services to proofread the manuscript. Certificate of language editing service is attached as **Annex 1**.

6. In conclusion, it would be beneficial if the authors could address the potential safety concerns associated with recommending Artesunate as a treatment for obesity. It is well known that Artesunate can cause severe side effects, such as renal failure, gastrointestinal issues, insomnia, neurological impairments, and fatigue, in a significant number of patients. Given the popularity of social media platforms for sharing diet and health advice, it is particularly important to carefully consider the potential risks and benefits of recommending Artesunate for obesity treatment. It would be in the interests of the authors and the journal Nature Communications to carefully communicate the results of this study and the potential risks associated with using Artesunate. It is important to note that Artesunate is not a safe medication for humans.

Response: Thank you for your feedback and we understand your concerns. Currently, artesunate is used as a first-line treatment for humans with severe malaria. Therefore, the side effects of artesunate reported are likely associated with the disease complication and severity. A recent study reported the usage of artesunate for treatment of severe malaria and found artesunate showed a good safety profile with rare and mild side effects. Only 8 out of 270 patients showed delayed haemolysis and all of them fully recovered later (Abanyie, Acharya et al. 2021).

We have referred and summarized the safety and toxicity report of artesunate submitted by European medicine agency in September 2021 (Procedure No. EMEA/H/C/005550/0000). The preclinical and clinical studies as described below indicate that artesunate is safe for human use if used in controlled dose.

- a) Toxicity assessment of artesunate was performed in rhesus monkey at a dose of 16, 32 and 128mg/kg/day for 7 days. Only few animals at a dose of 32mg/kg/day or higher displayed short-term neurological effect such as drooling and spontaneous motor activity but they **recovered back within an hour**. Brain autopsy of the rhesus monkey treated with artesunate upto the dose of 128mg/kg **showed no pathological changes**. Effect on gastrointestinal and hepatic function were only observed at a dose of 16mg/kg/day or higher. Apart from changes in the gastric pH at higher dose, **no changes were observed in the gut function** (muscle contraction and transit time).
- b) Telemetered beagle dogs were administered with artesunate at a dosage of 8mg/kg/day and **no changes were reported in the cardiovascular and respiratory functions**.
- c) A phase-1 clinical study used 2, 4 or 8mg/kg/ dose regimens for artesunate and observed mild to moderate reticulocytopenia and decreased reticulocyte count in healthy subjects but **it was reversed to baseline within of 8-10 days**. **No abnormalities were reported for liver functions**.
- d) A clinical study reported neurological defect in 10.5% patients undergoing artesunate treatment for severe malaria. However, nearly 8.8% patients **recovered and showed no defect at the time of discharge**.
- e) There was **no mortality reported** in artesunate-treated healthy controls in phase 1&2 clinical trials. Deaths were reported only in the patients with severe malaria (7 out of 102 patients), It was **not due to the artesunate administration** but because of severe malaria.

In conclusion, the artesunate dosages used in the study for the treatment of obesity in mice (20mg/kg) and cynomolgus monkey (6.4mg/kg) are much lower than the doses reported to induce adverse drug effects. We also analysed biochemical and hematological parameters in wild type and obese mice/monkeys with chronic artesunate treatment and found no marked abnormalities (**Supplementary table 1-2**). In addition, we found that artesunate did not induce nausea (**Supplementary figure 10a**) and gastrointestinal dysfunctions (**Supplementary figure 12**), two most common side effects of GLP-1-based therapeutics for obesity, in obese mice. Lastly, neurological changes were assessed by Morris water maze and no deficits were observed in artesunate-treated mice (**Supplementary figure 11**).

Reviewer #2

These authors seek to make the case that, based on pre-clinical data, artesunate is a candidate drug to treat obesity, these actions being mediated through induction of GDF15. We are told that artesunate is a drug licenced in the USA for treatment of malaria. The authors demonstrate that artesunate raises GDF15 levels in mice, from approximately 50 to 200 pg/ml at the end of a 12 day period of daily intraperitoneal administration. This rise in GDF15 levels is mediated by induction of the stress response with most induction occurring in liver. These findings and serum levels are comparable to those changes reported to be induced by metformin treatment in mice. The artesunate has no impact on body weight of lean mice, presumably because of the very small rise in serum levels. It does induce significant weight loss in obese mice and this is accompanied by reduction in food intake and improvement in glucose homeostasis and liver steatosis. The authors have used germ line deleted GDF15 receptor knock out mice and shown that the effects of artesunate on body weight do not occur in these GFRAL knock out obese mice. At this dose of GDF15 they could identify no sickness behaviour based on pica behaviour, but they have not used a more sensitive test in rodents such as condition taste diversion. Artesunate has comparable effects in spontaneously obese primates in whom there is also no emesis reported.

1. The actions of artesunate on weight loss are similar to those of metformin and liraglutide in the doses used within the experimental protocol. No information is provided on serum GDF15 levels in either the metformin or the liraglutide treated animals.

Response: Thank you for the important suggestion. Following the reviewer's suggestions, we examined the serum GDF15 levels in artesunate-, metformin- and liraglutide-treated obese mice. Despite no changes of GDF15 levels in liraglutide-treated mice, artesunate and metformin could significantly induce GDF15 upregulation in obese mice; this GDF15-inducing effect was strongest in the artesunate-treated group (result below).

This result was in line with the data showing that artesunate treatment led to the greater reductions in both body weight and food intake in obese mice when compared to metformin and liraglutide treatments (**Supplementary figure 4A-C**). Therefore, artesunate could be a potential anti-obesity drug, which certainly warrants further investigations with human clinical trials in the future.

2. The authors believe that this effect is largely mediated through actions on the mouse hind brain. Whilst this is probably true, the evidence they present for this is very sketchy and I would recommend that this part of the manuscript either be removed or substantially revised. If it is to be revised, photographs need to adequately display the whole or at least most of relevant hind brain sections, which require a much lower power view. The sections need to identify the anatomical regions in question and the exact regions in the brain that have been sampled, need to be identified. Highly selected powered views are not very useful in this instance, although they can be used as a supplementary source of information.

***Response:** To investigate whether the anti-obesity effect of artesunate is mediated by GDF15/GFRAL axis, we quantified the percentage abundance of active GFRAL neurons within the brainstem, using the neuronal activity marker cFOS. Following a bolus intraperitoneal dose of artesunate, wild-type mice showed robust activation of GFRAL+ neurons (**Figure 5G-H**). Instead, this inducing effect on neuronal activation was lost in *Gfral*^{-/-} mice (**Figure 5G-H**), suggesting that artesunate-mediated activation of neurons is primarily mediated by GDF15/GFRAL axis. In this revised data, we have shown the photographs with both **low and high magnifications** showing most of the relevant hind brain sections and provided the quantification for the changes of cFOS activation in GFRAL neurons.*

3. The authors have not sampled energy expenditure and physical activity in the treated rodents and this is a significant limitation. Based on the existing literature, the results of these studies are not likely to have a major impact on the outcome.

***Response:** This is an excellent suggestion. We have performed the experiment as advised. We found that, in addition to the suppression of appetite, artesunate maintains the energy expenditure in obese mice (**Supplementary figure 9B-C**). This observation is in line with the study showing that GDF15 encounters the compensatory reduction in energy expenditure associated with calorie restriction.*

4. This manuscript would be improved by improving the evidence that the effects of artesunate were GDF15 mediated. This would involve treating mice with recombinant GDF15 to raise serum levels to that reported within the manuscript and observe if similar weight loss results. This could be done by giving GDF15 via an infusion pump, which would mimic a much smoother changes in serum level seen with induction of GDF15 secretion by artesunate.

***Response:** Thank you for this important suggestion. To confirm the anti-obesity effect of artesunate is mediated through GDF15, we examined the effect of artesunate in AAV-mediated *Gdf15* knockdown mice. High fat diet- induced obese mice were injected with AAV expressing *Gdf15*-specific shRNA (*shGDF15*) to knock down GDF15 specifically in the liver in which artesunate-induced GDF15 is majorly derived from. We found that knockdown of hepatic GDF15 dramatically reduced the artesunate-induced elevation of GDF15 levels and abrogated the anti-obesity effect of artesunate (**Figure 4**), which was in alignment with the data showing that GFRAL is essential for the bodyweight-regulatory effect of artesunate (**Figure 5**).*

Due to the technical limitation, we unfortunately could not perform the experiment with infusion pump as suggested by the reviewer. Hopefully, our animal data with the validation in both GDF15 knockdown mice and GFRAL knockout mice can convincingly demonstrate the key role of GDF15-GFRAL signaling axis in artesunate effects.

5. Overall, the authors make a convincing case that artesunate is another in the growing list of compounds capable of inducing GDF-15 expression and raising its serum levels. It is not completely clear if this does or does not induce sickness behaviour in mice and this finding would be more convincing if they used conditioned taste aversion. However, this is a difficult area to evaluate in rodents and if it truly does fail to induce sickness behaviour, this is likely to be due to the modest rises in serum levels associated with the use of this compound or the slower rate of change in serum levels. It is possible that if used in humans to treat obesity, it might cause modest weight reduction, as is seen with metformin. Whether to determine if it has less gastrointestinal side effects in metformin, would require a human clinical trial to evaluate.

Response: *We performed the conditioned taste aversion test. Consistent with the data of pica behaviour, we found no changes in the preference to saccharin in artesunate-treated mice (Supplementary figure 10d).*

In addition, we also examined the gastrointestinal side effects in artesunate-treated normal and high fat diet-induced obese mice. We found no changes in the gastrointestinal transit and fecal output after artesunate treatment (Supplementary figure 12).

6. The referencing and background could do with some improvement. The primary references for GDF15's effect on body weight and obesity should be cited rather than random selected references. The issue of exercise, GDF15 and weight loss is controversial, with various publications showing differing results. Simply saying that induction of GDF15 in response to exercise is not sufficient to modulate food intake is misleading. This should either be discussed more fully or the sentence discussing this removed.

Response: *Thank you for your advice. We have included the primary reference which reported the role of GDF15-GFRAL signaling in regulating body weight and obesity (Reference no 9-14). We have removed the statement as advised and revised the manuscript.*

Reviewer #3

In this manuscript the Authors describe the effect of artesunate in for the treatment of obesity in in non-human primates. Non-human primates are important models for the understanding and treatment of obesity and related metabolic conditions including type 2 diabetes mellitus, the insulin-resistance metabolic syndrome and arterial hypertension. Previous extensive work by Folli F and collaborators have established mechanisms of insulin resistance and beta-cell failure in baboons.

The experiments in mice and non-human primates are well performed, and they point to GDF15 as an important mediator of satiety. Previous extensive Work by the groups of Martin Myers jr and Jens Bruning have provided major insights into the neuronal populations which are involved in the mechanisms regulating satiety and appetite.

What is lacking in the present work is how this GDF15 mediated pathways interacts with other neuronal subpopulations to regulate satiety. One way to start to uncover this could be performing triple immunostainings with leptin , MCH and other hormones with GDF15.

***Response:** Thank you for your insightful comments. In this study, we focused on identifying the molecular mechanism underlying the anti-obesity effect of artesunate. In order to achieve this research goal, we used different mouse models including AAV mediated GDF15 knockdown (Figure 4) and GFRAL knockout ($GFRAL^{-/-}$) (Figure 5) and confirmed that artesunate therapeutic effect was abolished in the loss of GFRAL-expressing neuronal populations, suggesting that GDF15/GFRAL signaling axis is essential for the bodyweight-lowering effect of artesunate.*

Leptin receptors ($LepRb$)-expressing neurons are mostly present in the nucleus tractus solitarius (NTS) of the brainstem region and regulates food intake by responses to leptin signal. Despite the functional similarity between GDF15 and leptin, a recent study showed that leptin-mediated anti-obesity effect is likely independent of GDF15-GFRAL signaling (Cheng, Ndoka et al. 2020).

Melanin concentrating hormone (MCH) are usually produced by the neurons present in the lateral hypothalamic area and their activation enhances the appetite food seeking behaviour in mice (Lord, Subramanian et al. 2021, Subramanian, Lauer et al. 2023). As the expression of GDF15 receptors are uniquely restricted to the area postrema region, GFRAL neurons unlikely co-express MCH. This is further supported by the single cell-transcriptome analyses of GFRAL neurons showing neither $LepRb$ nor MCH are co-expressed with GFRAL in the brainstem neurons (Zhang, Kaye et al. 2021).

Given the fact that leptin, MCH and GDF15 are circulating hormones and GDF15 is mainly produced in the peripheral tissues, it is unlikely that three hormones are co-expressed in the same cell population.

Previous studies have identified neuronal population in area postrema and nucleus tractus solitarius region for mediating the responses to GDF15 actions - “The cytokine GDF15 signals through a population of brainstem cholecystokinin neurons to mediate anorectic signalling” eLife 9:e55164. (Worth, Shoop et al. 2020)” It would be of interests to further investigate how brainstem

cholecystinin neurons mediate GDF15 signal in the future. However, it may be beyond the scope of this study.

To add power to the experiments in monkeys they should be performed with a cross-over design
Response: *This is an excellent suggestion. However, due to insufficient funding available, we unfortunately cannot compare the effect of metformin, liraglutide and artesunate on obese monkeys, which certainly deserves further investigations in the future.*

Annex 1

References

Abanyie, F., et al. (2021). "Safety and Effectiveness of Intravenous Artesunate for Treatment of Severe Malaria in the United States-April 2019 Through December 2020." Clin Infect Dis **73**(11): 1965-1972.

Bernasocchi, T., et al. (2021). "Dual functions of SPOP and ERG dictate androgen therapy responses in prostate cancer." Nat Commun **12**(1): 734.

Cheng, W., et al. (2020). "Leptin receptor-expressing nucleus tractus solitarius neurons suppress food intake independently of GLP1 in mice." JCI Insight **5**(7).

He, R., et al. (2022). "SULF2 enhances GDF15-SMAD axis to facilitate the initiation and progression of pancreatic cancer." Cancer Lett **538**: 215693.

Lord, M. N., et al. (2021). "Melanin-concentrating hormone and food intake control: Sites of action, peptide interactions, and appetite." Peptides **137**: 170476.

Myojin, Y., et al. (2021). "Hepatic Stellate Cells in Hepatocellular Carcinoma Promote Tumor Growth Via Growth Differentiation Factor 15 Production." Gastroenterology **160**(5): 1741-1754 e1716.

Subramanian, K. S., et al. (2023). "Hypothalamic melanin-concentrating hormone neurons integrate food-motivated appetitive and consummatory processes in rats." Nat Commun **14**(1): 1755.

Worth, A. A., et al. (2020). "The cytokine GDF15 signals through a population of brainstem cholecystinin neurons to mediate anorectic signalling." Elife **9**.

Zhang, C., et al. (2021). "Area Postrema Cell Types that Mediate Nausea-Associated Behaviors." Neuron **109**(3): 461-472 e465.

REVIEWER COMMENTS

Reviewer #1 (Remarks to the Author):

The authors have done a good job addressing my comments, but a few small concerns still remain.

1) For the AAV data, can the authors consistently write AAV8 and not just AAV throughout the text? AAV8 targets also kidney, where basal Gdf15 mRNA abundance and is increased by artesunate treatment. It's important that the authors show the levels of Gdf15 expression in liver and kidney for the AAV8-shGDF15 experiments

2) Fig S9 and Fig 4 describe the AAV8-shGDF15 experiments. Can the authors comment on why n = 6 in Fig S9 vs n = 5 in Fig 4.

3) I appreciate the safety discussion regarding artesunate. It was informative; however, it remains important to temper the potential for widespread usage. Long-term safety data in older individuals with overweight and obesity have not yet been obtained. Until such testing is conducted, one should refrain from trivializing the potential of self-medication with artesunate.

4) it's intriguing concerning the GDF15 antibody. We attempted to use the same antibody in our laboratory for liver and kidney samples in both WT and GDF15 KOs, but we did not observe specific bands. This disparity might be linked to specific Western blot conditions, such as the blocking solution used and the temperature at which the samples were boiled. Could the authors add as much detail as possible about the WB procedure to the manuscript. This could help the community.

Reviewer #2 (Remarks to the Author):

Whilst the authors have dealt with some of the issues i have raised, there are still some significant outstanding problems.

I'm still very troubled about the data and presentation of artesunate induced fos activation. The photographic presentation of the studied sections remains inadequate and does not conform to what would normally be expected in studies of the AP/NTS. Aside from this, in the figures presented (4G and

5G), the dotted lines are confusing ... are they to signify the area of the enlarged pictures? The authors describe neuronal activation in the NTS, however, pretty much all authors that have studied GDF15 induced neuronal activation in the hindbrain demonstrate neuronal activation dominantly in the area postrema with very small amounts of activation in the adjacent NTS which is not really what's described in this paper. If activation is mainly in the NTS its not likely to be mediated by GDF15 alone. There is no description of the amount of Artesunate that's been injected intraperitoneally nor is there any identification of the amount of time after injection that the mice were studied for fos induction. There is no description of the coordinates over which studies of fos induction was sampled. Quantification has been done as the number of fos positive cells per section is not optimal because it can be very difficult to accurately sample identical sections across the AP and NTS which is important. It's usually better to measure fos expression across the relevant nuclei or sub-section of the relevant nuclei in the hindbrain.

For these reasons I cannot change my earlier view as this section is unsatisfactory and needs substantial correction or deletion from the manuscript.

I note that the authors have added data on energy expenditure which helps round out the picture. However, as part of this total energy expenditure analysis, the data needs to be corrected for lean mass and it's then helpful to also study both light and dark phases separately as well as together. It's necessary to also present data on physical activity across the group as this can influence energy expenditure. Lastly, it can be helpful to look for carbohydrate and fat utilisation as weight loss can influence these parameters.

With respect of the lack of sickness behaviour seen with Artesunate administration, would the authors care to speculate as to whether this might be due to the fact that mice are exposed to steady stage GDF15 levels as opposed to pulsatile GDF15 levels as often occurs when recombinant GDF15 is administered to mice for experimental purposes.

As a general comment I believe that images of western blots should show the whole gel and not a highly selected part from the gel. In the case of a GDF15 this is a particular issue because aside from demonstrating the presence of nonspecific bands, GDF15 is not infrequently secreted in unsliced and partially sliced forms which differ in their molecular weights.

Reviewer #3 (Remarks to the Author):

I believe the Authors have addressed the Reviewers comments.

The manuscript lacks an appropriate study and referring to published literature in the field of molecular mechanisms of obesity (Jens Bruning and Martin Myers Jr) as well as in known very well studied models of obesity and diabetes in non-human primates, namely baboons (see a number of original manuscripts published by Franco Folli's group (Chavez.....Folli, Diabetes 2009; Guardado MendozaFolli PNAS 2011; Guardado MendozaFolli Diabetologia 2015; FiorentinoFolli JCI Insights 2019). Without these background papers, this paper does not achieve a publishable level for Nature Communications.

Response to Reviewers' comments

Reviewer #1

- 1) For the AAV data, can the authors consistently write AAV8 and not just AAV throughout the text? AAV8 targets also kidney, where basal Gdf15 mRNA abundance and is increased by artesunate treatment. It's important that the authors show the levels of Gdf15 expression in liver and kidney for the AAV8-shGDF15 experiments

Response: Thank you for the comment. We have revised the manuscript accordingly. According to your suggestion, we examined the expression of GDF15 in kidney tissues derived from mice with AAV8-shGDF15 or AAV8-shControl upon the treatment of artesunate by immunofluorescent staining and found that AAV8-shGDF15 indeed significantly reduced the kidney GDF15 expression (**Figure A below**). Along with the efficient knockdown of hepatic GDF15 (**Figure B below, also see the previous letter for the reviewer**), the artesunate-induced increase in circulating GDF15 was abrogated in DIO mice with AAV8-shGDF15 (**Figure S9A**).

A

Fig S9 and Fig 4 describe the AAV8-shGDF15 experiments. Can the authors comment on why n = 6 in Fig S9 vs n = 5 in Fig 4.

Response: *The sample sizes for the metabolism-related experiments including body weight, food intake, GTT, ITT and energy expenditure are same in both **Figure S9** and **Figure 4**. Only for immunofluorescence study (**Revised Figure S10-11**) we included n=5 because one of the samples was accidentally disrupted during sectioning and therefore it was excluded from the study.*

2) I appreciate the safety discussion regarding artesunate. It was informative; however, it remains important to temper the potential for widespread usage. Long-term safety data in older individuals with overweight and obesity have not yet been obtained. Until such testing is conducted, one should refrain from trivializing the potential of self-medication with artesunate.

Response: *Thank for the insightful comment. We agree that it's important to prevent the self-medication of artesunate by the readers until the clinical trials are performed with it. We have added a statement in the discussion as advised: "**further clinical trials are needed to validate the effectiveness and safety of artesunate for the treatment of obesity in humans.**"*

3) It's intriguing concerning the GDF15 antibody. We attempted to use the same antibody in our laboratory for liver and kidney samples in both WT and GDF15 KOs, but we did not observe specific bands. This disparity might be linked to specific Western blot conditions, such as the blocking solution used and the temperature at which the samples were boiled. Could the authors add as much detail as possible about the WB procedure to the manuscript. This could help the community.

Response: *Thanks for the suggestion. We have revised the methodology with more details.*

Reviewer #2 (Remarks to the Author):

- 1) Whilst the authors have dealt with some of the issues i have raised, there are still some significant outstanding problems. I'm still very troubled about the data and presentation of artesunate induced fos activation. The photographic presentation of the studied sections remains inadequate and does not conform to what would normally be expected in studies of the AP/NTS. Aside from this, in the figures presented (4G and 5G), the dotted lines are confusing ... are they to signify the area of the enlarged pictures? The authors describe neuronal activation in the NTS, however, pretty much all authors that have studied GDF15 induced neuronal activation in the hindbrain demonstrate neuronal activation dominantly in the area postrema with very small amounts of activation in the adjacent NTS which is not really what's described in this paper. If activation is mainly in the NTS its not likely to be mediated by GDF15 alone.

***Response:** Thank you for the valuable suggestions. To address this concern, we have revised the whole image dataset with new pictures which clearly display the area postrema (AP) region above the central canal (CC) (**Revised Fig S10-11**). We have also revised the figure legend accordingly. In addition to the activation of GFRAL neurons in AP region, previous studies have shown that GDF15 also activates a population of GFRAL^{-ve} neurons in both AP and NTS regions [1, 2], suggesting the interconnectivity and transactivation between GFRAL^{-ve} and GFRAL^{+ve} neurons. In alignment with these reported studies, our findings showed that GDF15 induced by ART treatment led to the activation of GFRAL⁺ neurons in the AP region and a small population of GFRAL⁻ neurons in the AP/NTS region as shown by cFos⁺ signal (**Fig S10-11**). However, the ART-induced neuronal activation was abolished in the Gfral^{+ve} neurons in both mice with GDF15 knockdown (shGDF15) (**Fig S10**) and the Gfral knockout mice (Gfral^{-/-}) (**Fig S11**). Consistently, depletion of GDF15 or Gfral knockout also abrogated the anti-obesity effect of ART in obese mice. This data clearly indicates that the bodyweight-lowering effect of ART is primarily mediated through GDF15/GFRAL signalling axis.*

- 2) There is no description of the amount of Artesunate that's been injected intraperitoneally nor is there any identification of the amount of time after injection that the mice were studied for fos induction.

***Response:** We are sorry for the misunderstanding. However, the time of sample harvest for studying the c-Fos activation was described in the figure legend (**Figure S10-11**) as follows - "Immunofluorescent staining of GFRAL (green) and c-Fos (red) in brainstem regions of brain sections 12 hours after intraperitoneal administration of ART (20 mg/kg) in wild-type (WT) and Gfral^{-/-} mice with DIO". We have also provided the information of ART dosage in the methodology section as well as in each of the figure legends.*

- 3) There is no description of the coordinates over which studies of fos induction was sampled. because it can be very difficult to accurately sample identical sections across the AP and NTS which is important. It's usually better to measure fos expression across the relevant nuclei or sub-section of the relevant nuclei in the hindbrain. For these reasons I cannot change my earlier view as this section is unsatisfactory and needs substantial correction or deletion from

the manuscript.

Response: Thank you for your feedback. We understand your concern and we have revised the methodology and figure legend with more methodology details.

We focused on the AP region for studying *c-Fos* induction in the *Gfral*^{+ve} neurons. Usually the AP region can be identified by visualizing the tissue sections for distinct anatomical landmarks, followed by the further confirmation with the morphological identification using mouse brain atlas [3-5]. We have adopted this approach to identify the AP region. To quantify the *cFos* expression in specific hindbrain neurons, *Gfral*^{+ve} cells that were *cFos*⁺ and *cFos*⁻ were counted in every third coronal section, thus at least 5 sections containing AP. The methodology we used in this paper to study AP region and *Gfral* neurons has been described in different published studies including our previous study [6] and other studies [4]. We hope the revised manuscript and the detailed explanation will address the concerns raised by the reviewer.

- 4) I note that the authors have added data on energy expenditure which helps round out the picture. However, as part of this total energy expenditure analysis, the data needs to be corrected for lean mass and it's then helpful to also study both light and dark phases separately as well as together. It's necessary to also present data on physical activity across the group as this can influence energy expenditure. Lastly, it can be helpful to look for carbohydrate and fat utilisation as weight loss can influence these parameters.

Response: Thank you for the important suggestion. We have added all the data accordingly (**Revised S9B-C, D-E**). The energy expenditure data has already been corrected for lean mass and we have revised it in the methodology section. Analyses of physical activity revealed that ART moderately increased the locomotor activity during the dark cycle, but this effect was abolished in mice with *shGDF15* (**Revised S9D-E**).

In addition, we analysed the potential effect of ART on carbohydrate and lipid metabolism in major metabolic tissues including liver, skeletal muscle (GCM) and iWAT from HFD-induced obese mice after artesunate treatment for two weeks by global metabolite profiling. Through LC-MS analyses, we did not find overt changes in the carbohydrate metabolism and therefore focused on comprehensive profiling of lipid metabolites. We found significant upregulation of prostaglandin D2 (PGD2), fatty acid esters of hydroxy fatty acids (FAHFAs) in liver and glycerophosphocholines (GPC) in muscle. These metabolites all have beneficial effects on the management of obesity.

PGD2, one of the major eicosanoids produced in the liver, is involved in different biological functions, such as body temperature regulation, hormone release and inhibition of pro-inflammatory responses [7]. FAHFAs are a group of fatty acids with reported anti-inflammatory and anti-diabetic effects [8-10]. A cohort study reported reduced levels of FAHFAs in obese conditions [11]. GPC is a derivative of choline and important precursor for the synthesis of neurotransmitters. Recent study reported the administration of GPC reduced body weight and food intake in diet-induced obese mice [12]. GPC has been shown to exert beneficial effects on obese/diabetic mice by reducing fat accumulation, adipogenicity, serum levels of triglycerides and total cholesterol [12, 13]. We also observed that fatty acids associated with obesity and insulin resistance [14] were reduced in the skeletal muscle and adipose tissue upon the artesunate

treatment, again revealing the beneficial effect of artesunate on bodyweight management and insulin sensitivity.

In this study, we demonstrated that the bodyweight-lowering effect of artesunate is primarily mediated via GDF15/GFRAL signalling axis. The modulating effect of artesunate on lipid metabolism as shown in our study is indeed in alignment with previous reports showing that GDF15 prevents obesity by increasing lipolysis. In the future, it would be of interest to investigate how GDF15/GFRAL modulates lipid metabolism, but this may go beyond the scope of this current study.

Please see the attached lipidomics data from liver, muscle and iWAT tissue.

5) With respect of the lack of sickness behaviour seen with Artesunate administration, would the authors care to speculate as to whether this might be due to the fact that mice are exposed to steady stage GDF15 levels as opposed to pulsatile GDF15 levels as often occurs when recombinant GDF15 is administered to mice for experimental purposes.

Response: Thank you for your suggestions. This is an interesting research question that certainly deserves further investigation in the future.

We postulate that there are two factors determining the anorectic effect of GDF15, including the rate of GDF15 elevation and the level of GDF15. Previous studies using transgenic mice with GDF15 overexpression showed that the stable increase in systemic GDF15 levels is effective to protect mice from high fat diet-induced obesity without the sign of sickness behaviours [15]. Under the feeding with normal diet, transgenic mice with GDF15 overexpression do not exhibit any overt phenotypes. Similar to our study, the GDF15/GFRAL-dependent anti-obesity effects of camptothecin and metformin also do not come with the anorexia and food aversion [16, 17]. These results suggest that unlike the rapid rise of GDF15 by the administration of recombinant GDF15, the steady rise of GDF15 induced by drugs does not induce sickness behaviour and is well-tolerated.

Concentration is another key factor for the adverse effect of GDF15. High levels of GDF15 ($>900\text{pg ml}^{-1}$) induced by chemo drug cisplatin has been shown to induce strong anorectic effects in animals [18, 19]. However, low levels of GDF15 ($<200\text{pg ml}^{-1}$) may not sufficient to suppress appetite and regulate bodyweight [17, 20]. In our study, we reported that artesunate treatment induced the elevation of GDF15 to around 400pg ml^{-1} in obese mice and 600pg ml^{-1} in obese non-human primates, both of which falls in the tolerated level. Therefore, artesunate promotes weight loss in both obese mice and monkeys without the induction of sickness behaviours.

However, to precisely study the difference between the steady upregulation of GDF15 and the pulsatile GDF15 levels, the technique of infusion pump is required and as mentioned in our previous response letter our laboratory is not equipped with the required instrument for this experiment at the moment. We hope that our intensive experiments with multiple genetic knockout models and non-human primates can convince you that anti-obesity effect of artesunate is primarily mediated with GDF15/GFRAL signalling axis and does not induce sickness behaviour. We also hope that the technical limitation would not compromise your confidence in our study.

- 6) As a general comment I believe that images of western blots should show the whole gel and not a highly selected part from the gel. In the case of a GDF15 this is a particular issue because aside from demonstrating the presence of nonspecific bands, GDF15 is not infrequently secreted in unsliced and partially sliced forms which differ in their molecular weights.

Response: We have included all the unprocessed blots in the supplementary figure (**Figure S15**). To probe different protein targets with different molecular weights in the same membrane, we have sliced the membrane in accordance to the molecular sizes of our target proteins, which is also a common practice for western blotting. Moreover, the specificity of the GDF15 antibody used in this study has been validated in the GDF15 knockdown experiment as shown in the previous response letter for the reviewer (pls. also see the figure below).

Reviewer #3 (Remarks to the Author):

- 1) I believe the Authors have addressed the Reviewers comments. The manuscript lacks an appropriate study and referring to published literature in the field of molecular mechanisms of obesity (Jens Bruning and Martin Myers Jr) as well as in known very well studied models of obesity and diabetes in non-human primates, namely baboons (see a number of original manuscripts published by Franco Folli's group (Chavez.....Folli, Diabetes 2009; Guardado MendozaFolli PNAS 2011; Guardado MendozaFolli Diabetologia 2015; FiorentinoFolli JCI Insights 2019). Without these background papers, this paper does not achieve a publishable level for Nature Communications.

Response: Thank you for the insightful comment and feedback. We agree that inclusion of these classic papers in the manuscript will make the manuscript more informative and interesting for the reader. We have added the relevant citations accordingly, Pls see (reference no - 4, 5, 10, 17, 20).

References –

1. Worth, A.A., et al., *The cytokine GDF15 signals through a population of brainstem cholecystinin neurons to mediate anorectic signalling*. Elife, 2020. **9**.
2. Hsu, J.Y., et al., *Non-homeostatic body weight regulation through a brainstem-restricted receptor for GDF15*. Nature, 2017. **550**(7675): p. 255-259.
3. Matsushita, T., et al., *Sustained microglial activation in the area postrema of collagen-induced arthritis mice*. Arthritis Res Ther, 2021. **23**(1): p. 273.
4. Zhang, C., et al., *Area Postrema Cell Types that Mediate Nausea-Associated Behaviors*. Neuron, 2021. **109**(3): p. 461-472 e5.
5. Paxinos, G. and K.B. Franklin, *The mouse brain in stereotaxic coordinates: compact*. 2004, Amsterdam.
6. Chow, C.F.W., et al., *Body weight regulation via MT1-MMP-mediated cleavage of GFRAL*. Nat Metab, 2022. **4**(2): p. 203-212.
7. Joo, M. and R.T. Sadikot, *PGD synthase and PGD2 in immune resposne*. Mediators Inflamm, 2012. **2012**: p. 503128.
8. Lee, J., et al., *Branched Fatty Acid Esters of Hydroxy Fatty Acids (FAHFAs) Protect against Colitis by Regulating Gut Innate and Adaptive Immune Responses*. J Biol Chem, 2016. **291**(42): p. 22207-22217.
9. Kuda, O., et al., *Erratum. Docosahexaenoic Acid-Derived Fatty Acid Esters of Hydroxy Fatty Acids (FAHFAs) With Anti-inflammatory Properties*. Diabetes 2016;65:2580-2590. Diabetes, 2016. **65**(11): p. 3516.
10. Dongoran, R.A., et al., *Determination of Major Endogenous FAHFAs in Healthy Human Circulation: The Correlations with Several Circulating Cardiovascular-Related Biomarkers and Anti-Inflammatory Effects on RAW 264.7 Cells*. Biomolecules, 2020. **10**(12).
11. Kellerer, T., et al., *Fatty Acid Esters of Hydroxy Fatty Acids (FAHFAs) Are Associated With Diet, BMI, and Age*. Front Nutr, 2021. **8**: p. 691401.

12. Kim, G.W. and S.H. Chung, *The beneficial effect of glycerophosphocholine to local fat accumulation: a comparative study with phosphatidylcholine and aminophylline*. Korean J Physiol Pharmacol, 2021. **25**(4): p. 333-339.
13. Izu, H., et al., *Anti-diabetic effect of S-adenosylmethionine and alpha-glycerophosphocholine in KK-A(y) mice*. Biosci Biotechnol Biochem, 2019. **83**(4): p. 747-750.
14. Boden, G., *Obesity and free fatty acids*. Endocrinol Metab Clin North Am, 2008. **37**(3): p. 635-46, viii-ix.
15. Macia, L., et al., *Macrophage inhibitory cytokine 1 (MIC-1/GDF15) decreases food intake, body weight and improves glucose tolerance in mice on normal & obesogenic diets*. PLoS One, 2012. **7**(4): p. e34868.
16. Lu, J.F., et al., *Camptothecin effectively treats obesity in mice through GDF15 induction*. PLoS Biol, 2022. **20**(2): p. e3001517.
17. Coll, A.P., et al., *GDF15 mediates the effects of metformin on body weight and energy balance*. Nature, 2020. **578**(7795): p. 444-448.
18. Breen, D.M., et al., *GDF-15 Neutralization Alleviates Platinum-Based Chemotherapy-Induced Emesis, Anorexia, and Weight Loss in Mice and Nonhuman Primates*. Cell Metab, 2020. **32**(6): p. 938-950 e6.
19. Borner, T., et al., *GDF15 Induces Anorexia through Nausea and Emesis*. Cell Metab, 2020. **31**(2): p. 351-362 e5.
20. Klein, A.B., et al., *Pharmacological but not physiological GDF15 suppresses feeding and the motivation to exercise*. Nat Commun, 2021. **12**(1): p. 1041.

REVIEWER COMMENTS

Reviewer #1 (Remarks to the Author):

The authors have largely addressed my concerns however I request that the immunofluorescent staining that found that AAV8-shGDF15 significantly reduced the kidney GDF15 expression and the WB showing partial knockdown of hepatic GDF15 are included in the manuscript.

Reviewer #4 (Remarks to the Author):

The authors have made substantial efforts to address the comments from reviewer #2 resulting in notable improvements to the manuscript. However, some data-related concerns still persist.

1. Regarding the neuronal data, the author has responded to reviewer #2's feedback by replacing the images, focusing primarily on the AP rather than the NTS regions, and relocating the image to the supplemental figures. Nonetheless, several issues persist with the images:

a) In Figure s10, it appears that the image used for "shControl+Vehicle" was also employed for "shGDF15+Vehicle," albeit with some adjustments in exposure and size. It is reasonable to assume that these treatments would have been administered to different mice, and therefore, the images should not be identical.

b) GFRAL and FOS expression typically exhibit non-uniform distribution throughout the entire AP/NTS region. I would recommend that the author include the coordinates within the image (e.g., bregma xyz) or in the legend, as previously suggested. In Figures s10 and s11, it is evident that the representative images selected from different treatments do not closely match their anatomical position.

c) Has there been a re-analysis of the quantification of FOS+/GFRAL+ cells in the AP? The graph appears to utilize the same data as the previous version, in which quantification was conducted both in the AP and NTS. As previously mentioned, given the non-uniform expression of GFRAL and FOS throughout the entire AP, quantification of GFRAL+FOS+ cells should be performed in sub-regions, with matched serial sections spanning the entire AP.

2. The lipidomic data is intriguing, and I concur with the authors that presenting this data falls beyond the scope of the current study. Nevertheless, it would be beneficial if the author could include data on the respiratory exchange rate as an indicator of whether Artesunate exhibits a preferential effect on carbohydrate and fat utilisation as sources of energy.

Response to reviewers' comments

Reviewer #1

- 1) The authors have largely addressed my concerns however I request that the immunofluorescent staining that found that AAV8-shGDF15 significantly reduced the kidney GDF15 expression and the WB showing partial knockdown of hepatic GDF15 are included in the manuscript.

Response: Thanks for the suggestion. We have added the data in supplementary figures (Revised Figure S9A-B).

Reviewer #4

The authors have made substantial efforts to address the comments from reviewer #2 resulting in notable improvements to the manuscript. However, some data-related concerns still persist.

- 1) Regarding the neuronal data, the author has responded to reviewer #2's feedback by replacing the images, focusing primarily on the AP rather than the NTS regions, and relocating the image to the supplemental figures. Nonetheless, several issues persist with the images:
 - a) In Figure s10, it appears that the image used for "shControl+Vehicle" was also employed for "shGDF15+Vehicle," albeit with some adjustments in exposure and size. It is reasonable to assume that these treatments would have been administered to different mice, and therefore, the images should not be identical.

Response: Thank you for pointing out the careless mistake in figure preparation and we sincerely apologize for it. We have revised the figure with the correct representative images for shControl + Vehicle group (Revised Figure S11A).

- b) GFRAL and FOS expression typically exhibit non-uniform distribution throughout the entire AP/NTS region. I would recommend that the author include the coordinates within the image (e.g., bregma xyz) or in the legend, as previously suggested. In Figures s10 and s11, it is evident that the representative images selected from different treatments do not closely match their anatomical position.

Response: In each representative image, we have included the distance from the bregma (Revised Figure S11A, S12A). We have also added the information in the figure legends and methodology section.

- c) Has there been a re-analysis of the quantification of FOS+/GFRAL+ cells in the AP? The graph appears to utilize the same data as the previous version, in which quantification was conducted both in the AP and NTS. As previously mentioned, given the non-uniform expression of GFRAL and FOS throughout the entire AP, quantification of GFRAL+FOS+ cells should be performed in sub-regions, with matched serial sections spanning the entire AP.

***Response:** Thanks you for the feedback and we are sorry for the misunderstanding. We have included the quantification of FOS+/GFRAL+ cells confined to the AP region (**Revised Figure S11B, S12B**). Coronal sections within -7.48 to -7.64mm distance from the bregma were examined for the quantification. Every third serial section from each mice/group was quantified in order to span the entire AP region.*

2. The lipidomic data is intriguing, and I concur with the authors that presenting this data falls beyond the scope of the current study. Nevertheless, it would be beneficial if the author could include data on the respiratory exchange rate as an indicator of whether Artesunate exhibits a preferential effect on carbohydrate and fat utilisation as sources of energy.

***Response:** This is an excellent suggestion. We have added the respiratory exchange rate (RER) in the **revised Figure S10E-F**. We found that artesunate moderately reduced the RER, indicative of increased fat utilisation and reduced carbohydrate metabolism.*

REVIEWER COMMENTS

Reviewer #1 (Remarks to the Author):

My requests were addressed.

Reviewer #4 (Remarks to the Author):

The authors have addressed my concerns. However, with regards to S12B, quantifying GFRAL+/cFOS+ seems illogical when GFRAL has been deleted. Perhaps the authors should consider quantifying cFOS+ as a measure of overall neuronal activation influenced by the absence of GFRAL.

Response to reviewers' comments

Reviewer #1

My requests were addressed.

Response: Thank you very much for your efforts.

Reviewer #4

The authors have addressed my concerns.

Response: Thank you very much for your efforts on reviewing our manuscript.

With regards to S12B, quantifying GFRAL+/cFOS+ seems illogical when GFRAL has been deleted. Perhaps the authors should consider quantifying cFOS+ as a measure of overall neuronal activation influenced by the absence of GFRAL.

Response: Thank you for your suggestions. We have revised the quantification of cFos⁺ cells accordingly (Figure S12B).